# A mutualistic interaction between *Streptomyces* bacteria, strawberry plants and pollinating bees

Da-Ran Kim[1], Gyeongjun Cho[2], Chang-Wook Jeon[2], David M. Weller[3], Linda S. Thomashow[3], Timothy C. Paulitz[3] & Youn-Sig Kwak [1,2]*

Microbes can establish mutualistic interactions with plants and insects. Here we track the movement of an endophytic strain of *Streptomyces* bacteria throughout a managed strawberry ecosystem. We show that a *Streptomyces* isolate found in the rhizosphere and on flowers protects both the plant and pollinating honeybees from pathogens (phytopathogenic fungus *Botrytis cinerea* and pathogenic bacteria, respectively). The pollinators can transfer the *Streptomyces* bacteria among flowers and plants, and *Streptomyces* can move into the plant vascular bundle from the flowers and from the rhizosphere. Our results present a tripartite mutualism between *Streptomyces*, plant and pollinator partners.

[1] Department of Plant Medicine and Institute of Agriculture and Life Science, Gyeongsang National University, Jinju 52828, Republic of Korea. [2] Division of Applied Life Science and Research Institute of Life Science, Gyeongsang National University, Jinju 52828, Republic of Korea. [3] US Department of Agriculture, Agricultural Research Service, Wheat Health, Genetics, and Quality Research Unit, Pullman, WA 99164-6430, USA. *email: kwak@gnu.ac.kr

Microbes colonize both the surface and the interior of plants in highly adapted communities essential to the growth, development, and health of their hosts[1,2]. The microbial consortia differ in composition between above- and below-ground plant parts depending on the chemical, physical and environmental characteristics of the habitat[3,4]. Collectively, the microbes encode a second genome that expands the physiological function of the host by producing growth hormones, fixing nitrogen, mobilizing nutrients, inducing resistance mechanisms and defending against biotic and abiotic stresses[5]. Modern microbe-plant interaction concepts[1,6] recognize these unique mutualistic relationships as evolutionary units[7–9] with capabilities that exceed those of either partner alone[10].

Plant-associated microbial consortia differ according to the tissues they inhabit, and the selection, assembly and interactions with the host and other members of the microbial community in which they reside have only recently begun to be dissected[11–13]. The soil is an important source of much of the microflora found in both the rhizosphere and phyllosphere[14,15]. Flowers are essential for the biological success of most plants, yet flowers are the plant parts most vulnerable to biotic and abiotic stresses[16,17]. Microbial communities of this ephemeral habitat remain poorly understood, and the interactions among beneficial and deleterious microbes that colonize flowers are especially elusive.

Plants rely on members of their microbiota to contribute early defense mechanisms against pathogen attack, especially by necrotrophic pathogens[18–20], but whether or how beneficial agents travel within or among plants is less clear. In this research, we used a managed strawberry ecosystem to trace a plant-mutualistic microbe partnership that has expanded to include a pollinator in its interactions. The mutualistic agent, an antibiotic-producing strain of *Streptomyces*, moves within the plant vascular system from the roots and from flowers, and is carried to other plants by pollinator bees. The *Streptomyces* defends the plant against attack by the gray mold pathogen *Botrytis cinerea* and protects the bees against insect pathogens. Using the plant and the pollinator ecosystems, *Streptomyces* benefits by effective dispersion.

## Results

### Gray mold disease incidence and microbial assemblages.
From November, 2013 to March, 2014 we collected samples at 2-week intervals from strawberry (cv. Meahyang) cultivated in high-bed greenhouses in Jinju, Republic of Korea (Supplementary Fig. 1a–f). Sequence analyses of bacterial DNA from flowers and pollen samples (Supplementary Tables 1 and 2), identified operational taxonomic unit (OTU) numbers ranging from 10 to 30 for flower and from 14 to 25 for pollen samples, respectively. Thus, regardless of the origin and sampling time, the communities were relatively simple (Fig. 1a, b). Heatmaps and cluster analysis indicated that in both the flower and the pollen samples, communities shifted over time and differentiated into high diversity (HD, weeks 0–12) and low diversity (LD, weeks 14–24) groups over the course of the season (Fig. 1c–e). Comparing the OTUs in the HD and LD groups from flowers, Proteobacteria were the most abundant, followed by Actinobacteria. An OTU identical to the sequence of *Streptomyces globisporus* NRRL B-3872 (accession number, EF178686) was the most frequently detected (Supplementary Table 3). Abundant bacteria in the pollen HD group included mainly Proteobacteria, with fewer Actinobacteria and Firmicutes (Supplementary Table 4). Among the combined flower and pollen samples, OTUs of Streptomycetaceae were consistently represented only in the HD group (Supplementary Table 5). Actinobacteria comprised 44.9% of the total bacteria in flowers and 43.8% in pollen (Supplementary Table 6). As previous reports have hypothesized that the microbial

community can benefit plant health[21,22], we next established the relationship between the microbiota and the occurrence of gray mold disease and spore density of the pathogen *Botrytis cinerea* in the air during strawberry growth (Supplementary Fig. 2a–e). Gray mold is the most important disease of many greenhouse-grown crops including strawberries. In most cases, the gray mold pathogen is endemic in commercial greenhouses, where it infects and causes gray mold disease of strawberry. Disease incidence increased dramatically on flowers at week 16 and peaked at week 20, when 33% of all berries were damaged (Supplementary Fig. 2e). Comparing the pattern of disease incidence with that of the bacterial community structure throughout the growing season revealed that gray mold occurrence increased rapidly at week 14 when the bacterial community shifted from the HD to the LD configuration lacking the *S. globisporus*. The decline in the *Streptomyces* OTU population beginning at week 14 correlated with a sharp increase in symptoms of gray mold disease (Fig. 1g). Among common OTUs in the HD group of both flower and pollen samples, the most abundant OTU was identical to the sequence of *S. globisporus* NRRL B-3872 (Fig. 1f) and prompted the hypothesis that if the bacteria contribute to the maintenance of plant health, there would be specific beneficial strains present.

### Selection and identification of mutualistic isolates.
Totals of 887 and 324 isolates were recovered from flowers and pollen, respectively. After antifungal screening in vitro, 44 isolates from flowers (Supplementary Table 7) and 22 from pollen (Supplementary Table 8) were selected as putative beneficial strains. Among these, *Streptomyces* species were targeted based on the results of sequencing, bacterial community shifts and gray mold disease incidence. Seven *Streptomyces* strains from flowers and nine from pollen (Supplementary Table 9) were identified as *S. badius*, *S. globisporus*, or *S. griseus* by sequencing 16S rRNA (99% similarity). Multigene phylogenies of 16S rRNA, *gyrB* and *recA* genes of strains SP6C4 and SF7B6 showed these bacteria to be related most closely to *S. griseus* (Fig. 2a). Recently, *S. badius*, *S. griseus*, and *S. globisporus* have been categorized as part of the "*S. griseus* group"[23]. Strains SP6C4 (isolated from pollen at week 10) and SF7B6 (isolated from flowers at week 12) were indistinguishable from each other, although they were recovered from different sources and sampling time points to minimize the probability of recovering the same clone twice. Strains SP6C4 and SF7B6 showed strong antifungal activity against *B. cinerea* as well as colonization activity on flowers (Fig. 3a, b). Comparison of the two genome sequences by Average Nucleotide Identity (ANI) revealed 99.99% sequence identity and unexpectedly, both shared 99.99% genomic sequence identity with strain *S. griseus* S4-7 (Fig. 2b), the well-characterized mutualistic microbe isolated from the strawberry rhizosphere in 2012 that protects strawberry against Fusarium wilt disease[19]. All three strains also shared the same sequence maps by the Web Artemis Comparison Tool (ACT) (Fig. 2c). *S. globisporus* C1027 showed 96.1% genomic sequence similarity as the closest related reference bacterium among the three strains (Fig. 2d). Bacteria with >94% similarity in ANI are considered to be the same species[24]. Accordingly, the three strains were re-classified as *S. globisporus* S4-7, SF7B6, and SP6C4.

### *S. globisporus* SP6C4 movement in planta.
Identity among the three strains also suggested that they could colonize flowers and even pollen attached to bee bodies, leading us to hypothesize that the bacteria establish in the rhizosphere, move upward internally as endophytes, and can transfer to pollinators. This model is consistent with a mutualistic relationship in an ecosystem that includes both plants and insects. The hypothesis was tested in two

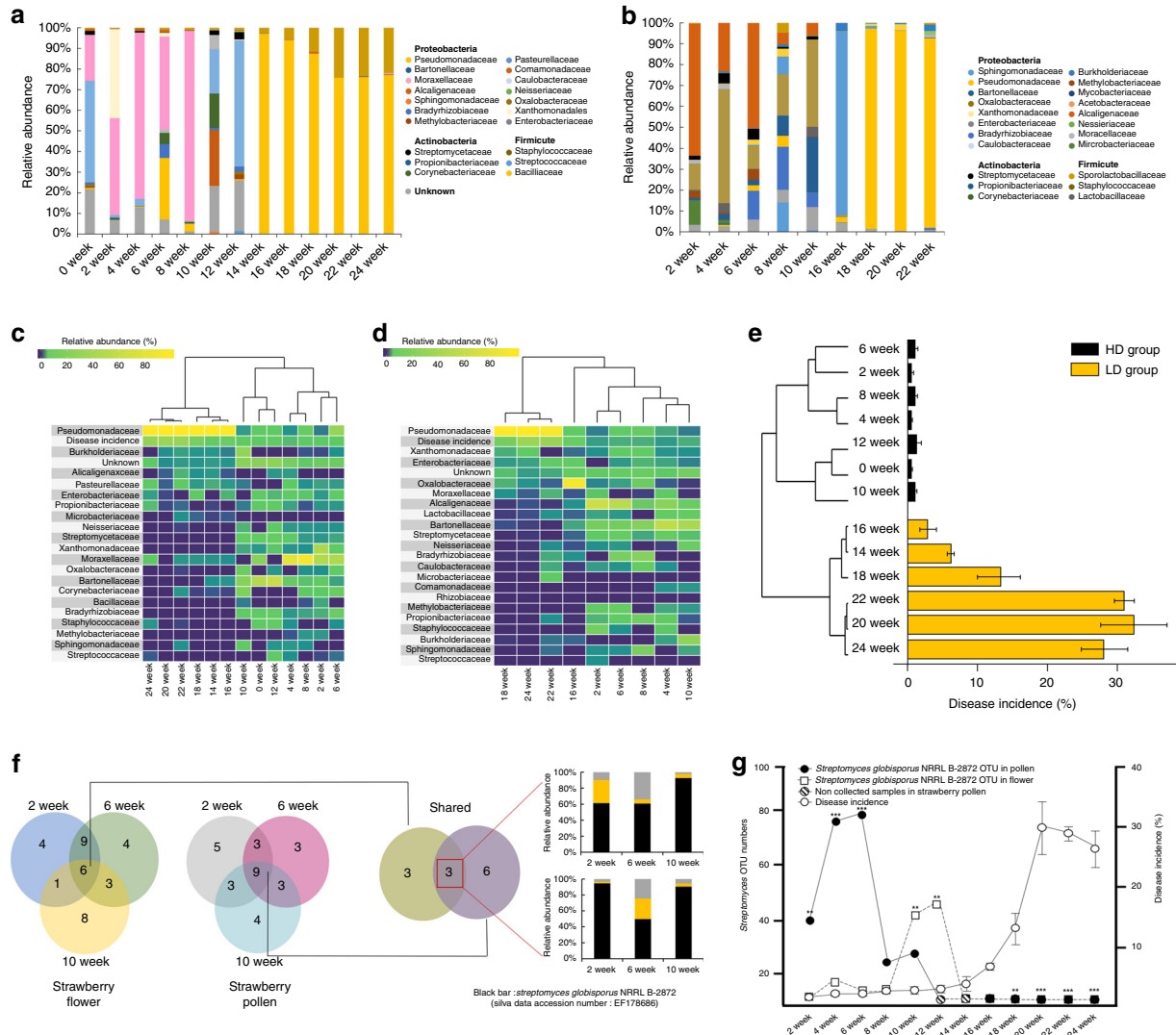

**Fig. 1** Microbial diversity of strawberry flowers and pollen. **a** Pyrosequencing of microbes in strawberry flowers ($n = 9$, 13 independent experiments) and **b** pollen ($n = 2$, 9 independent experiments). Taxonomic assignment was conducted at the family level with the Silva database (http://www.arb-silva.de/) and a cutoff of 97% similarity. Flower and pollen samples were collected from November 2013 (0 week) to March 2014 (24 week). Heatmap of hierarchical clustering of bacterial communities by 16S rRNA region. **c** Flower samples and **d** pollen samples. Heatmap color (purple to yellow) displayed from low to high abundance of each OTU. **e** Beta diversity tree (Minkowski distance) of samples with gray mold disease incidence. **f** Venn diagram of common OTU numbers in flower and pollen samples during the period of low gray mold disease incidence. **g** Gray mold incidence over a growing season as related to *Streptomyces* OTU read numbers. Gray mold incidence, bars represent standard error of nine blocks, each block contains 150 plants. Star (*) indicates statistically significant differences between disease incidence and OTU numbers of *Streptomyces globisporus* NRRL B-2872, which is identical to SP6C4 and SF7B6 by *t*-test (*P* value < 0.05). Bars represent standard error. **a**, **b**, **e**, **f**, **g** Source data are provided as a Source Data file

separate treatments conducted by either dipping roots of strawberry into a suspension of strain SP6C4 ($10^6$ cfu/mL) or by dropping 100 µL of the bacterial culture by pipette onto flower stamens and carpels. Both treatments included two independent experiments, one in which the introduced strain was detected with the strain-specific marker gene *lanM*[25] and the other in which it was labeled with mCherry. All stem, crown and root tissues were surface-sterilized prior to fluorescence microscopy and analysis by qPCR for *lanM*. Five days after root dipping, the bacteria were detected in the endosphere of root tissue by qPCR (Fig. 4a, b). By 10 days, mCherry fluorescence was seen in root and stem vascular bundle tissues (Fig. 4c) which, like those harboring the wild-type strain, tested positive for *lanM* and maintained population sizes of up to $10^6$ *lanM* copy/g of tissue until 20 days (Fig. 4b, c). At 30 days, both mCherry fluorescence and

population sizes tested by qPCR were greatly reduced in the xylem of the stem (Fig. 4b, d).

After flower inoculation, the mCherry-labeled strain was observed on the stigma and pistil of flowers (Fig. 4f, h). The bacteria retained a population size of at least $10^5$ *lanM* copy/g of tissue and mCherry fluorescence was observed along the filament tissues until 15 days (Fig. 4e, g, i). The bacteria were initially detected up to $10^5$ *lanM* copy/g of tissue in both the stem at 10 days and the crown at 15 days, where high population densities were maintained for up to 30 days (Fig. 4e). In cross sections just above the crown, the middle part of the stem and the petiole, mCherry fluorescence was detected in vascular bundles from 10 days (Fig. 4j) up to 30 days (Fig. 4k). These results provide evidence that SP6C4 can translocate endophytically from both rhizosphere and flowers to stem vascular bundles.

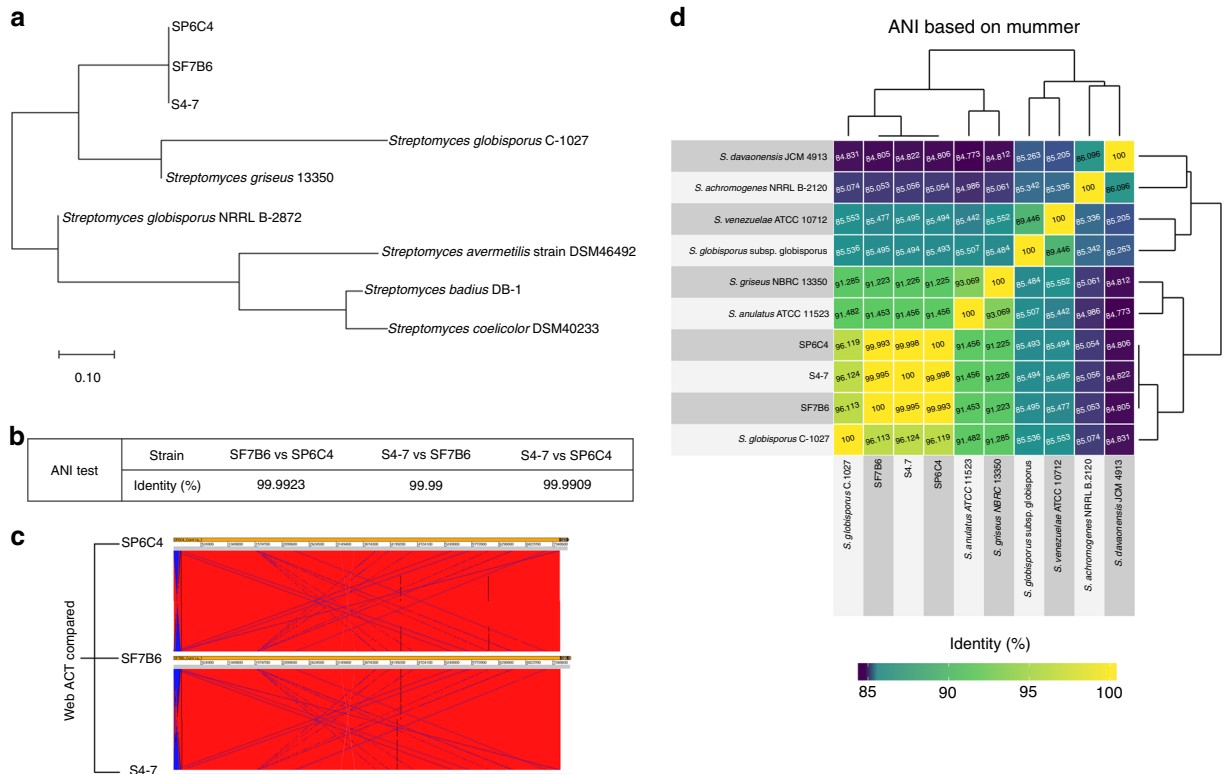

**Fig. 2** Identification of strains SP6C4, SF7B6, and S4-7 at the genome level. **a** Phylogenetic tree of strains SP6C4, SF7B6, and S4-7 based on 16S rRNA and housekeeping genes (*gyrB* and *recA*). Sequences were compared by multiple alignment analyses using the Maximum Likelihood method of MEGA 7. **b** Average nucleotide identity (ANI) of the strains. **c** Web ACT (http://www.webact.org/WebACT), blast parameters of nucleotide mismatch penalty −1; gap opening penalty 1; gap extension penalty 2. **d** ANI genome alignment threshold was raised to 94%. **b**–**d** Source data are provided as a Source Data file

To further evaluate movement from plant to plant by a pollinator, three sets of flowers were prepared in a cage. The first set of flowers received suspensions of a hygromycin-marked strain on the surface of flower stamens and carpels, the second set received no bacterial treatment and the third set of flowers also had no bacteria and was covered with a cap to prevent honeybee access (Fig. 5a). Five days later, surface populations on the inoculated flowers were up to $10^7$ cfu/g of flower, consistent with successful colonization of the flower surface. At the same time, strain SP6C4 was detected at $10^5$ cfu/g of honeybee body and at $10^6$ cfu/g of flower on non-inoculated flowers visited by the honeybees. However, flowers that were protected from bee access by a cap showed no SP6C4 (Fig. 5b). In addition, the strain colonized the gut of the honeybee (Fig. 3c, d). Collectively, these findings indicate that the strain can move from one plant to another via transport by the pollinator.

**Gray mold disease suppression by SP6C4 in the phyllosphere**. We next addressed why the plant and the honeybees act as vehicles of transport of the strain. To investigate the effect of the translocated *Streptomyces* on the incidence of gray mold disease in the phyllosphere, an experiment was conducted from January until March, 2016, in which strain SP6C4 or a water control was sprayed four times, from week 2 to week 8, on strawberry plants in a greenhouse with a replicated complete block design (Supplementary Fig. 3b, c; Supplementary Tables 10 and 11). At week 10, 31% of the flowers sprayed with water were infected with gray mold (Fig. 5e), but the disease incidence of the SP6C4-sprayed plants was 12% (Fig. 5d). Actinobacteria on the SP6C4-sprayed and control flowers were present at relative abundances of 2% or nondetectable levels (Fig. 5d, e). Populations of SP6C4 were detected at a density of $10^4$ *lanM* copy/g of flower immediately

after introducing the strain at week 2, began to increase at week 6, and reached $10^6$ *lanM* copy/g at week 10 (Supplementary Fig. 4a). In contrast, on control plants, the population of the strain remained at basal levels by qPCR (Supplementary Fig. 4b). Gray mold spore density remained at levels of ≤log 0.9 cfu/L of air throughout the period of the experiment in the greenhouse (Fig. 5f).

To evaluate whether strain SP6C4 vectored by honeybees could influence the severity of gray mold disease, another experiment was established from November 2016 to February 2017. SP6C4 was introduced into a greenhouse via a bee-vectoring system set up in front of the hive (Supplementary Fig. 3d; Supplementary Tables 10 and 11) from weeks 4 to 8. In the control greenhouse, symptoms of gray mold disease first appeared at week 4 and by week 10, 42% of the flowers were diseased (Supplementary Fig. 4e). In contrast, disease incidence in the greenhouse in which SP6C4 was delivered by the bee-vectoring system remained at 12% through week 10 (Supplementary Fig. 4e). In the treated greenhouse, SP6C4 was detected at $10^4$ *lanM* copy/g of flower immediately after introduction at week 4, and the population reached $10^6$ *lanM* copy/g of flower at week 10 (Supplementary Fig. 4c). In contrast, the population in the control greenhouse remained at basal levels by qPCR (Supplementary Fig. 4d). Thus, in both this and the experiment above in which SP6C4 was applied by spraying, the incidence of gray mold disease was reduced by the *Streptomyces* treatment.

**Protecting the pollinator against entomopathogens**. Bees also benefited from the interaction with *Streptomyces* because strain SP6C4 inhibited the entomopathogens *Paenibacillus larvae* and *Serratia marcescens* (Fig. 5c). Caged bees fed pollen containing no bacteria or only strain SP6C4 for 5 days had 28% mortality,

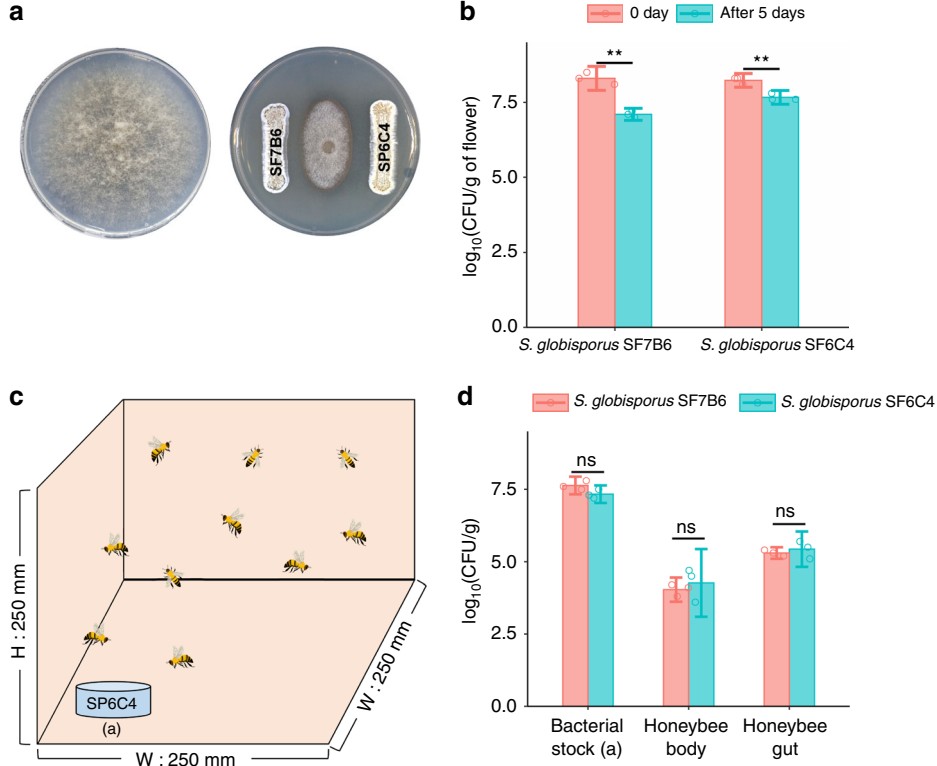

**Fig. 3** SP6C4 antagonism in vitro against gray mold pathogen, colonization of the flower surface and the honeybee body. **a** Mycelium growth inhibition of *Botrytis cinerea* by strain SP6C4 and SF7B6 on PDK agar. Two strains were streaked on PDK agar with toothpicks. After 3 days, the pathogen agar block (0.7-mm) was placed at the center of the plate, then the plate was incubated for 7 days at 28 °C. **b** Flowers ($n = 3$) received 100 µL of the bacterial suspension ($10^6$ cfu/mL, 0.1% methyl cellulose). The flowers were incubated in a growth chamber (16 h light, 28 °C; 8 h of dark, 24 °C). After 5 days, they were collected into 30 mL of PBS buffer and spread on PDK media with hygromycin (80 µg/mL). Bars represent standard deviation (Paired sample *t*-test: SF 0 day vs SF after 5 days, $P = 0.002854$; SP 0 days vs SP after 5 days, $P = 0.003858$; **$P < 0.01$). **c** Experimental illustration of SP6C4 colonization of the honeybee body and gut. **d** SP6C4 or SF7B6 ($10^6$ cfu/mL, 0.1% methyl cellulose) were introduced in a 5-cm Petri dish. After 5 days, bacterial population density was investigated both on the surface and in the gut of the honeybees. For honeybee surface colonization by the bacteria, the bodies were placed in 50 mL of PBS buffer and sonicated for 10 min. For gut samples, the bodies were rinsed three times with sterile water and gut tissue was extracted. The samples were diluted to $10^{-7}$, spread on PDK agar with hygromycin (80 µg/mL), the plate was incubated at 28 °C for 5 days and colony forming units were calculated. Differences in the bacterial colonization among the treatments were analyzed by independent two sample *t*-test (Bacterial stock: $P = 0.007393$, Honeybee body: $P = 0.5705$, Honeybee gut: $P = 0.5355$ and bars represent standard deviation. **d** Source data are provided as a Source Data file

whereas those fed pollen inoculated with bee pathogens had 73% (*P. larvae*) or 88% (*S. marcescens*) mortality, respectively. Mortality was reduced ~50% when each pathogen was fed in a mixture with SP6C4, indicating that it reduced mortality even when fed in mixtures with the entomopathogens. These and the findings above demonstrate that the microbial strain was capable of providing protection against disease to both the plant and the insect.

## Discussion

Beneficial plant-microbe interactions in the rhizosphere have been recognized for well over a century[26], but the diversity, magnitude and consequences of these associations remain continually evolving areas of investigation[1,27]. Similarly, whereas plant-microbe relationships have often been characterized in relation to disease, it is becoming increasingly apparent that the outcome of such relationships is driven in part by rhizosphere mutualists either supportive or antagonistic to pathogens[4,21,28]. The recruitment and maintenance over time of such specific, environmentally selected mutualists supportive of host health is an integral part of the phytobiome concept[14,28,29], and the functions mediated by these strains can explain in part the unique community structure of different plant species[4] or even

interconnected plant parts. In our study, the absence of the *Streptomyces* mutualist was coincident with lack of suppression of the *Botrytis* pathogen during later sampling periods, and its dominance in flowers and pollen during early sampling periods, provides support for the suggestion of Durán et al.[22] that individual members of the microbial community can have a central role in the health of the host plant.

We previously reported that a strain of *Streptomyces* from disease suppressive field soil became a dominant member of the bacterial population and protected strawberry against *Fusarium oxysporum* f. sp. *fragariae* during mono-culture[19]. Building on these results, we have shown that a genetically identical *Streptomyces* strain from the rhizosphere of greenhouse-grown plants enters the endosphere and moves upward to deliver protection against an air-borne foliar pathogen. Only a fraction of the rhizosphere community has the capacity to colonize the plant endosphere in a process actively controlled by the host[3,30], but once established, these bacteria can travel via the xylem to aboveground tissues[31]. Our evidence that SP6C4 (accession numbers in Supplementary Table 12) is transferred from the rhizosphere to the endosphere, as well as from the flower to pollinators, offers adapted microflora not only the potential for dispersion within the plant, but also efficient translocation to other plants that subsequently may benefit as the *Streptomycetes* move systemically

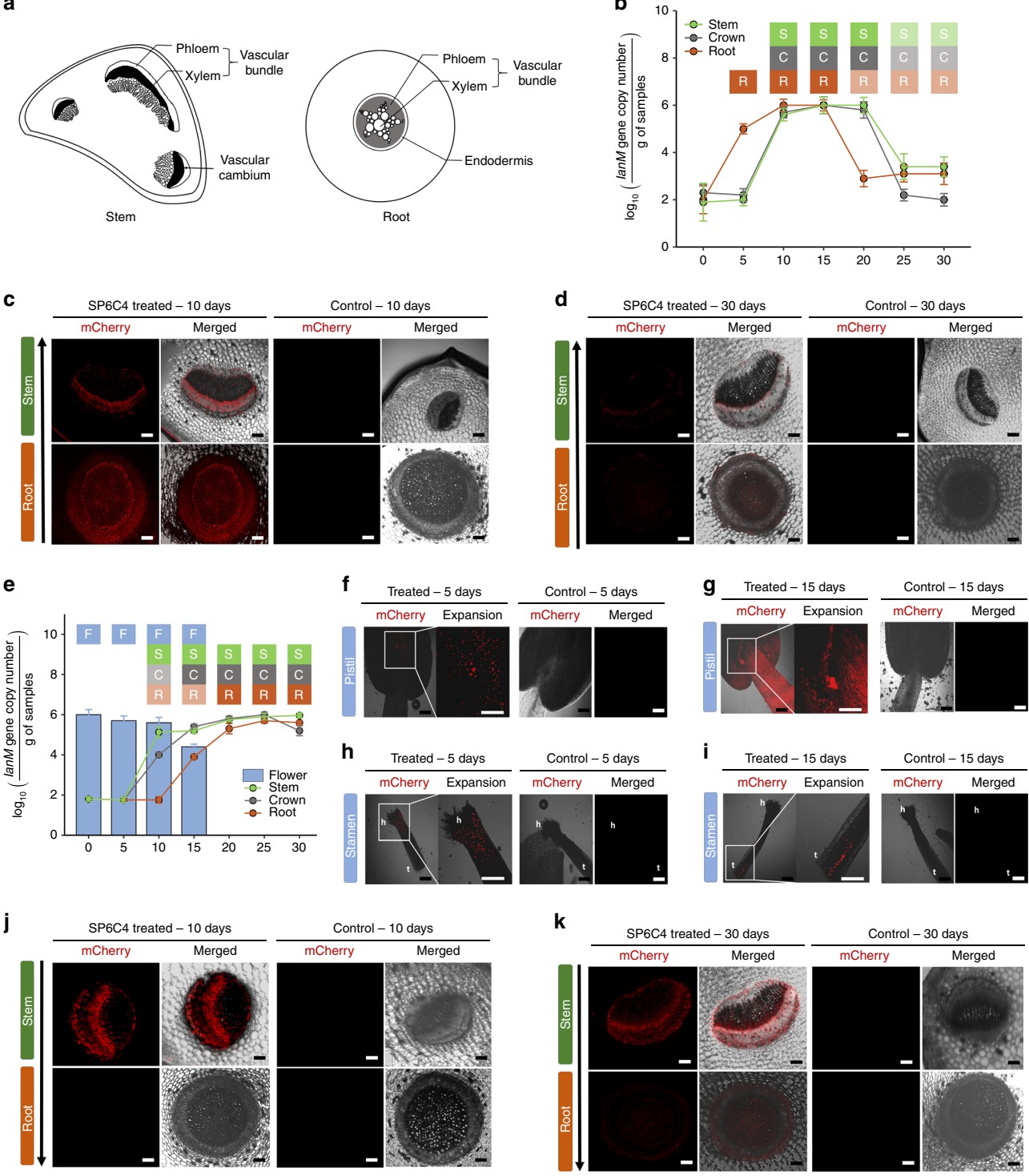

**Fig. 4** Translocation of the *S. globisporus* SP6C4 from below- to above-ground and vice versa. **a** Illustration of the cross section of a strawberry stem and root. **b** *S. globisporus* SP6C4 was introduced onto the root surface in a suspension with 0.1% methyl cellulose. *S. globisporus* SP6C4 was detected with a *lanM*-specific primer using qPCR SYBR Green® TOYOBO master mix. Bars represent ± SE (*n* = 3, 3 independent experiments). **c** Confocal microscopy of plant samples sectioned at 10 days. **d** 30 days after inoculation. Fluorescence of strain SP6C4-mCherry (530–560 nm detection channels) showing translocation from the rhizosphere to above-ground tissue. **e** Strain SP6C4 introduced on the flower and detected by *lanM*-specific qPCR. Movement of the strain from above- to below-ground plant parts. Bars represent ± SE (flower: *n* = 5, 3 independent experiments; root and stem: *n* = 3, 3 independent experiments; crown: *n* = 3, 3 independent experiments). **f** SP6C4-mCherry on the pistil at 5 days (mCherry channel scale bar, 100 µm), 15 days samples (**g**). **h** SP6C4-mCherry on stamen at 5 days (**i**) and 15 days stem and root at 15 days (**j**). **k** 30 days after inoculation. The plant was maintained in a growth chamber with 16 h light, 28 °C, 45% relative humidity, and 8 h dark, 24 °C, 45% relative humidity. **b**, **e** Source data are provided as a Source Data file

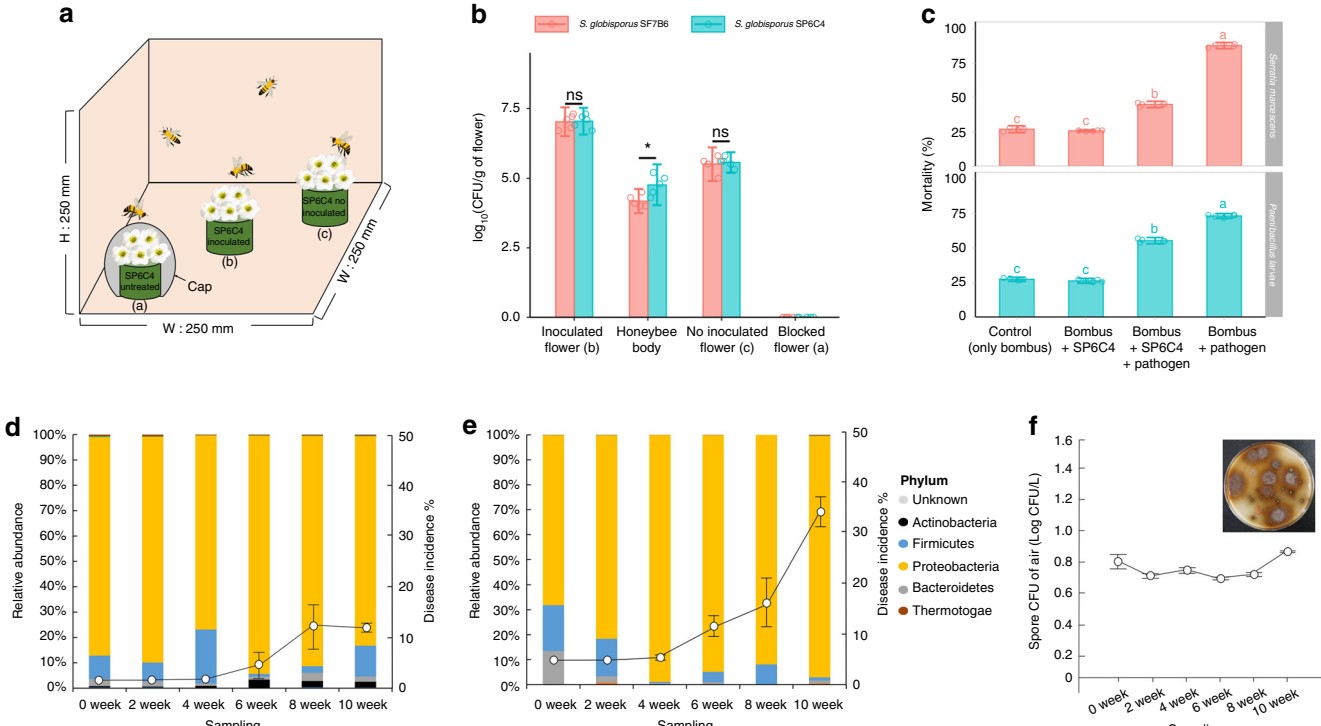

**Fig. 5** *S. globisporus* SP6C4 dispersed from flower to flower, reduced insect mortality and incidence of gray mold. **a** Illustration of honeybee transport of SP6C4 from plant to plant. (a): SP6C4 untreated and honeybee access blocked by a cap, (b): flower stamens and carpels inoculated with a suspension (100 μL) of SP6C4 ($10^6$ cfu/mL), (c): SP6C4 untreated flower ($n = 5$, 3 independent experiments). **b** Population densities of SF7B6 and SP6C4 on flowers and honeybee bodies. Surface bacteria were detached by sonication for 30 mins at 35 kHz. *Streptomyces* strain was isolated on PDK media amended with hygromycin B (80 μg/mL). SF7B6 and SP6C4 treatments were analyzed by independent two sample *t*-test, *$P < 0.05$; inoculated flower: $P = 0.9025$, honeybee body: $P = 0.0212$, no inoculated flower: $P = 0.7141$. Bars represent standard deviation. **c** Mortality of *Bombus impatiens*. Entomopathogens ($10^5$ cfu/mL) were added with or without SP6C4 ($10^6$ cfu/mL) to pollen. Bars with different letters are significantly different according to Duncan's test and ANOVA (*Serratia marcescens*: $P = 2e-16$; *Paenibacillus larvae*: $P = 2e-16$). Bars represent standard deviation. **d** Strain SP6C4 ($10^7$ cfu/mL with 0.1% methyl cellulose) was sprayed or **e**, not sprayed. The bacteria were sprayed by a sprayer (HP-2010, Korea; 1.5 L discharge capacity per min) and gray mold disease incidence was counted in three replicate plots. Length of each block was 15 meters ($n = 450$ plants per block). For analysis of the microbial community, flower samples were collected from January 2016 (0 week) to March 2016 (10 week). Taxonomy data were converted to Newick format on R (version 3.4.0) script. The converted taxonomic data were visualized with ggtree (version 3.4.11), ggplot2 (version 3.2.0), and agricolae (version 1.2–9) packages. **f** *Botrytis cinerea* spores in air collected by a spore trap every other week with BSTM media and the medium was incubated at 28 °C for 5 days to calculate the spore density. Bars represent standard error. **b**, **c**, **d**–**f** Source data are provided as a Source Data file

throughout them. As the vector in this process, the pollinator gains protection against common entomopathogens[32] but the flower serves as the hub in this tripartite system that benefits all three partners. While flowers typically harbor less diversity and fewer inhabitants than rhizosphere or phyllosphere communities[14,33], our results may provide evidence that selected members of the anthosphere may have a greater role in the plant ecosphere than previously has been recognized.

To better understand the persistence of microorganisms in the environment requires detailed knowledge of their individual capabilities and needs as well as recognition of the give-and-take involved in their interactions with other members of the community. The relationships among the microbe, the plant, and the insect in this study exemplify Commoner's laws of ecology[34]: that "everything is connected to everything else" within an interaction, with members interdependent. We suggest that in our study, the plant served as a host in the space inhabited by the three partners, the insect provided plant-to-plant transport and the *Streptomyces* maintained the partnership by protecting against pathogens that attack the plant and the insect. From an ecological perspective, when the soil, the plants and the pollinators are considered as separate environments inhabited by microorganisms, the *Streptomyces* in our system transformed the three into a mutualistic continuum with the plants. Our findings present mutualistic

interactions among individual members, but natural or agricultural ecosystems have many more complex and sophisticated elements. A better understanding of the relationships within such mutualistic phytobiomes will facilitate the use of the native microbiota and specific introduced agents to defend against biotic and abiotic stresses, thus more sustainably protecting the world's food supply.

## Methods

**Sampling and pyrosequencing of flowers and pollen.** Flowers and pollen were collected for microbial community analysis at 2-week intervals from November, 2013 to March, 2014 from strawberry (cv. Meahyang) cultivated in high-bed greenhouses in Jinju, Republic of Korea. At each sampling time, a replicate sample of 10–15 flowers was randomly collected from among 150 plants present in each of the nine replicate plots located at the edges and central rows of the greenhouse (Supplementary Fig. 1a–c). Each sample was placed in different 50-mL Falcon tubes, yielding nine tubes per greenhouse. Pollen also was recovered during strawberry growth from a collector attached to a beehive in each of two greenhouses (Supplementary Fig. 1d–f). Flower and pollen samples were stored in a portable refrigerator after collection and transported to the laboratory within 30 min, where each sample was weighed and adjusted to 1 g. For flower samples, 30 mL of 1× Phosphate-Buffered Saline (PBS) buffer (10× PBS: 8 g of NaCl, 0.2 g of KCl, 1.44 g of $Na_2HPO_4$, 0.24 g of $KH_2PO_4$, adjusted to a pH of 7.4 and a final volume of 1 L) was added. Pollen samples received 20 mL of 1× PBS buffer. Then the sample tubes were sonicated at 35 kHz (20 min, 4 °C) to detach microbes and 3 mL of the mixture was centrifuged (12,470 × *g*, 10 min, 4 °C). The supernatant solution was discarded and total DNA in the pellet was purified using a genomic

DNA isolation kit (Solgent, Daejeon, Korea). For each sampling time, DNA was extracted individually from nine flower samples and two pollen samples, and then the DNA of the flower or pollen samples was pooled (20 μL from each sample, DNA concentrations ranging from 180 to 200 ng/μL). To check the quality of the DNA and to determine the composition of the communities, 16S rRNA PCR reactions were conducted. Total DNA (100 ng) was reacted with the primer pair 27 mF (5′-gagtttgatcmtggctcag-3′) and 518R (5′-wttaccgcggctgctgg-3′) to amplify the V1–V3 region. Pyrosequencing was performed using 454 GS-FLX titanium system with a Roche Genome Sequencer (GS) FLX software (v 3.0). Samples were placed on a PicoTiter plate to obtain raw signals of sequences, and after being filtered to remove background noise the signals were sorted by barcode tag sequences. Sequencing reads were checked both for nucleotide quality scores (average Phred score >20) and read lengths (>300 bp). Rarefaction was calculated with iNEXT (version 2.0.12) in R (version 3.4.4). Operational taxonomic units (OTUs) were established using CD-HIT-OTU clusters, rRNA tags, and Mothur (version 1.33.0). Nucleotide sequence similarity of 16S rRNA sequences was followed at similarity levels: phylum, >75%; class, >80%; order, >85%; family, >90%; genus, >94% and species, >97%. The microbial communities, top OTUs and heatmaps displaying clustering were conducted with the R package. GenBank accession numbers are presented in Supplementary Table 12.

**_Botrytis cinerea_ spore density and incidence of gray mold**. A spore sampler (UCK Bio-Culture TMU Pump, USA) (Supplementary Fig. 3a) and Botrytis Spore Trap Medium (BSTM: glucose 2 g, NaNO₃ 0.1 g, K₂HPO₂ 0.1 g, MgSO₄·7H₂O 0.2 g, KCl 0.1 g, chloramphenicol 0.2 g, pentachloronitrobenzene 0.02 g, 80% manganese ethylene bisdithiocarbamate 0.02 g, 12% fenarimol 0.1 mL, tannic acid 5 g, agar 20 g per 1 L, pH 4.5) were used to trap conidia of the pathogen[35]. The BSTM plates were inserted into the sampler and air was aspirated over the plates (240 L, 2 min), which were then sealed with Parafilm and incubated at 28 °C for 5 days. Colony numbers of the pathogen were counted and spore densities were calculated and expressed in log colony forming units (cfu)/L of air.

Gray mold disease incidence and spores of the pathogen were collected at 2-week intervals in the greenhouses. Disease symptoms appeared as brown spots on petals, leaves and fruits, with masses of gray or brownish spores eventually produced on infected stems, or as mummified berries (Supplementary Fig. 2a–d). Differences in disease incidences among the treatments were analyzed by ANOVA followed by the Turkey HSD ($P = 0.05$) for mean separation.

**Isolation and antifungal activity of _Streptomyces_ strains**. Bacteria isolated from flowers and pollen in 1× PBS solution were diluted $10^{-8}$-fold in sterile water, plated in triplicate onto tryptic soy agar (TSA: tryptic soy broth (Difco) 30 g, agar 20 g per 1 L), cultured at 28 °C for 10 days, and CFUs were calculated based on the number of colonies in the terminal dilution plates. Each colony was inoculated into 100 μL of tryptic soy broth (TSB) in a 96-well plate, which was shaken at 28 °C for 2 days. Then, 100 μL of 50% glycerol was mixed into the wells by pipetting and the plates were sealed with Platemax® pierceable aluminum sealing film (Axygen, USA) and stored at −80 °C.

Antifungal _Streptomyces_ isolates in the bacterial collections were detected by a three-step screening procedure. First, bacteria from the 96-well plates were replica-plated onto Omni-trays (Sigma-Aldrich, USA) of PDK medium (potato dextrose (Difco) 10 g, peptone 10 g, agar 20 g per 1 L) and cultured at 28 °C for 2 days. Then, 4-mm-diam. plugs of actively growing mycelia of _B. cinerea_ were placed between the bacterial colonies. Second, colonies that inhibited fungal growth were suspended in sterile water, the concentration was adjusted to $10^6$ cfu/mL, and 10 μL was spotted onto PDK agar and streaked along a 3-cm-long line. Three days later, a 4-mm-plug of the fungus was placed at the center of the plate, 2 cm from the bacteria. Fungal inhibition was evaluated 5 days later. This screening was repeated with selected inhibitory isolates, with antifungal activity defined based on the distance between the pathogen and the antagonist after 5 days and denoted as − no inhibition; + (low inhibition), 0.1–0.5 cm; + + (medium inhibition), 0.5–1 cm; + + + (strong inhibition), 1–1.5 cm; + + + + (the highest inhibition), >1.5 cm.

**Sequencing of 16S rRNA and housekeeping genes**. Two independent strains, SP6C4 and SF7B6, were grown in TSB amended with 20% sucrose, transferred to MS medium (soya flour, 20 g; mannitol, 20 g; agar, 20 g per 1 L water)[36] and cultured at 30 °C for 7 days. Spores were collected into 1.5-mL Eppendorf tubes and genomic DNA was extracted by using a CTAB lysozyme procedure[37]. DNA quality was determined with a NanoDrop 2000C spectrophotometer (Thermo, Denmark). For identification of the isolates, full-length 16S rRNA, _gyrB_, and _recA_ genes were sequenced. PCR was performed with 1 μL of DNA, 4 μL of dNTPs (2 mM), 1 μL of 10 pmol of each primer (Supplementary Table 13), 0.3 μL of KOD FX Neo (1.0 unit/μL), 20 μL of 2X PCR buffer, and sterile water to a final volume of 40 μL. The PCR program consisted of an initial denaturation at 98 °C for 5 min, 30 cycles at 98 °C for 30 s, 55 °C for 30 s, 72 °C (2 min for 16S RNA and _gyrB_; 1 min for _recA_), and a final extension at 72 °C for 10 min. Amplicons were visualized in 1% agarose to confirm the size (1.4-kb for 16S rRNA, 1305-bp for _gyrB_, 913-bp for _recA_)[38,39] and sequenced by the Sanger method at Solgent (Daejeon, Korea). Sequences were aligned using ClustalW (http://www.ebi.ac.uk/Tools/msa/clustalo)[40] and phylogenetic trees were created by MEGA 7 software with a Maximum Likelihood algorithm[41].

**PacBio genome sequencing with _Streptomyces_ strains**. Genomes were sequenced with the Pacific Biosciences RSII (PacBio, USA) single molecule real-time (SMRT) system and non-hybrid hierarchical genome (HGAP version 2.3) pre-assembly Quiver (http://www.pacificbiosciences.com/devnet/). Whole-genome alignment used the Web ACT program (Artemis Comparison Tool, Center for Bioinformatics). Circular genome mapping and analysis were conducted with BLAST Ring Image Generator (BRIG) v 0.95 (http://sourceforge.net/projects/brig/)[42]. RAST version 2.0 release 70 with genetic code option 11 and other automatic pipeline options and SEED-based prokaryotic genome public databases were employed to annotate[43] the genomes. Results were run with the cluster finder algorithm, a minimum cluster size CDS of 5, minimum cluster finder probability score of 0.6, and automatic remaining options at the site.

**Colonization of flowers and movement by pollinators**. To monitor _Streptomyces_ colonization of flowers, a single colony was streaked onto MS medium and incubated for 10 days at 28 °C. Mature spores were collected in 1 mL of distilled water with a sterile cotton ball, filtered to remove bacterial mycelia, and adjusted to $10^9$ cfu/mL. This spore stock was inoculated into TSB broth amended with 20% glucose and 1% mannitol and incubated in a shaking incubator (120 rpm) at 28 °C for 10 days. The bacteria ($10^6$ cfu/mL) then were diluted with 1% methyl cellulose (final concentration of 0.1%) and 100 μL was dropped onto flowers (5 replications; each replication, $n = 3$ flowers). The flowers were incubated in a growth chamber (16 h light, 28 °C, 45% relative humidity; 8 h of dark, 24 °C, 45% relative humidity). Each treatment received 100 mL of water daily to maintain humidity. After 5 days, flowers (1 g) were harvested in 30 mL of PBS buffer and homogenized for 10 min. The samples were serially diluted in sterilized water, then spread on PDK media containing hygromycin (80 μg/mL). The plates were incubated at 28 °C for 5 days and bacterial densities were calculated for each sample (Supplementary Fig. 3b).

To assess _Streptomyces_ colonization of the honeybee, 7 mL of SP6C4 ($10^6$ cfu/mL in 0.1% methyl cellulose) in a Petri dish (5 cm diam.) and honeybees ($n = 10$) were released into a plastic cage ($25 \times 25 \times 25$ cm, 3 replications; Supplementary Fig. 3c). After 5 days, the bacterial population density was investigated both on the surface and in the gut of the honeybees. For colonization of the body, each bee was placed in 50 mL of 1× PBS buffer and sonicated for 15 min. For gut samples, the bodies were rinsed three times with sterile water and gut tissue was extracted with a Zeiss Stemi 508 stereo microscope (Zeiss, Germany) and homogenized in 1 mL of PBS buffer. Buffer samples were then serially diluted, spread on PDK medium with hygromycin (80 μg/mL), and incubated at 28 °C for 5 days.

To determine whether _Streptomyces_ strains could be moved among flowers by pollinators, three sets of flowers were established in a plastic cage ($25 \times 25 \times 25$ cm) (Fig. 4a). The first set received no bacterial treatment and was covered with a transparent plastic cap to prevent access by the bees (_Apis mellifera_). The second set also received no bacteria but had no plastic cap, which allowed access to the bees. Flowers in the third set were inoculated with 100 μL ($10^6$ cfu/mL) of a hygromycin-resistant derivative of strain SP6C4 suspended in 0.1% methyl cellulose, and the cups were not capped. Next, five _A. mellifera_ bees were released into the cage, and flower samples were harvested 5 days later, at which time those visited by bees had become withered. The population density of SP6C4 on the bee bodies and the flowers of each treatment was determined. A 1 g sample of flowers or a single bee was placed in 50-mL Falcon tubes containing 30 mL (flower) or 20 mL (bee) of PBS buffer. Tubes were sonicated at 35 kHz with cooling for 10 min, the buffer solution was filtered with two layers of cheesecloth, serially diluted, vortexed, and spread onto PDK medium amended with 80 μg/mL hygromycin. The plates were incubated at 28 °C for 5 days and population sizes were calculated for each sample. This experiment was conducted with three independent biological replications and all results were statistically analyzed by independent two sample _t_-test.

**Bacterial movement between the rhizosphere and the flower**. To observe bacterial movement from below- to above-ground plant parts, 45-day-old strawberry roots were dipped for 5 min into a suspension of strain SP6C4 at $10^6$ cfu/mL grown in PDK broth and amended to 0.1% methyl cellulose. The inoculated plants were sown in plastic pots (9 cm diam.) containing 450 g of commercial potting mix and the pot was covered with plastic wrap to prevent bacterial spread by watering or airflow and then incubated in a growth chamber (16 h light, 28 °C, 45% relative humidity; 8 h of dark, 24 °C, 45% relative humidity).

For movement from the flower to the rhizosphere, 100 μL of a culture of SP6C4 grown to $10^6$ cfu/mL in PDK broth containing 0.1% methyl cellulose was dropped by pipette onto petals. Root- and flower-inoculated plants were harvested at 0, 5, 10, 15, 20, 25, and 30 days after inoculation. At each sampling day, three plants were removed from the pots. Tissue (root, crown, and stem) samples were cut into 2-cm-long pieces and placed in 50-mL Falcon tubes. For surface sterilization, the pieces were washed in 70% ethanol, 1% NaOCl for 1 min, and finally rinsed five times with sterile water. The sterilized plant pieces were transferred into new 50-mL Falcon tubes with PBS buffer (30 mL for flower; 20 mL for stem, crown and root samples) and the pieces were sonicated at 35 kHz with cooling for 30 min and dried on sterilized filter paper for 15 min in a fume hood. The dried samples (root, stem) were re-cut 0.8-cm-long from each end to obtain the middle parts (0.5-cm-long). The segments were homogenized in 1 mL PBS and then sonicated at 35 kHz with cooling for 10 min. The pre-sterilized crown samples were cut into cubes

$(0.5 \times 0.5 \times 0.5$ cm) and homogenized in 5 mL PBS. Flower samples followed the same sterilization process and dried flower pieces were homogenized in 500 μL PBS. Each sample (500 μL PBS) was mixed with an equal volume of CTAB and DNA was extracted. Strain movement was detected by qPCR with *lanM* as the strain-specific marker gene, which previously had been verified as a S4-7-specific detection marker[25]. For the qPCR reaction, 100 ng of DNA was used as template, with 25 μL of SYBR Green® TOYOBO master mix (QPK-201T, Japan), 1 μL of each forward and reverse primer, and 16 μL of HPLC grade $H_2O$. The PCR program included initial denaturation at 98 °C for 1 min, denaturation at 98 °C for 30 s, followed by annealing at 59 °C for 30 s, and elongation at 72 °C 45 s for 40 cycles. qPCR was performed with CFX Connect™ Optics Module Real-Time PCR System (Bio-Rad, USA). The experiment was conducted with five biological replications and qPCR had 15 replications for each tissue and time point sample.

**Fluorescent imaging of SP6C4 in planta**. To visualize bacterial movement, mCherry fluorescence of SP6C4 and confocal microscopy were used; this experiment was performed independently of the qPCR detection experiment. The mCherry gene in pET21a(+)-HIStag-mCherry (plasmid #70719, Addgene, Cambridge, MA, USA) was employed to generate a fluorescent derivative of strain SP6C4. The mCherry region was amplified with pBRrevBam-PacI (5′-ttaattaaggt-gatgtcggcgatatagg-3′) and the T7 terminator primer (5′-gctagttattgctcagcgg-3′) as an 807-bp fragment, ligated into pGEM–T easy vector (Promega, USA), and transformed into *E. coli* DH5α (Cat#18365017, Thermo Fisher, Waltham, MA, USA). Transformants were selected on LB agar and the plasmids were recovered with a Dokdo Mini-prep Kit (ELPIS-Biotech, Korea) and digested with *Xho*I and *Pac*I. At the same time, pIJ10257 was cut with the same enzymes, and the digested plasmid and the mCherry fragments were purified from agarose gels and ligated together by T4 ligase. The construction was transformed into *E. coli* ET12567/ puZ8002. Conjugation with SP6C4 was by the same procedure as in the muta-genesis protocol. After conjugation, colonies were selected on agar containing hygromycin (80 μg/mL) and by detection of a hygromycin resistance gene by PCR with primers hyg-det 3: 5′-tccgctgtgacacaagaatc-3′ and hyg-det 5: 5′-cggctcatcac-caggtagg-3′. Confocal microscopy was performed with a Zeiss LSM 510 laser scanning microscope (Zeiss, Germany). To confirm labeling with mCherry, 50 μL of a spore stock of the fluorescent SP6C4 strain was prepared on sterilized cov-erslips inserted at a 45° angle into MS medium and grown for 4 days at 30 °C. Fluorescence was detected at 488 nm excitation and 530–560 nm detection chan-nels, respectively.

For visualization of movement from the rhizosphere to above-ground tissue, the root was dipped into the mCherry-labeled strain suspension ($10^6$ cfu/mL, 0.1% methyl cellulose) prepared as described above in the movement assay by qPCR. To treat the anthosphere, a strawberry plant with flowers was planted in a plastic pot (90 mm diam.), and 100 μL of the diluted bacterial suspension ($10^6$ cfu/mL) in 0.1% methyl cellulose was applied to the petal of a flower. Plants and soil were covered with plastic wrap to prevent contamination. Rhizosphere, root and stem samples were collected at 0, 5, 10, 15, 20, 25, and 30 days after inoculation. Anthosphere samples were collected at intervals of 5 days for 15 days, after which the flowers had withered and become detached. Tissue samples were cut into cross sections using sterile scissors and fixed with 4% paraformaldehyde. All samples were prepared for microscopic observation with the ClearSee protocol[44].

**SP6C4 treatment to reduce strawberry gray mold**. To test whether strain SP6C4 reduces gray mold disease incidence on strawberry, the bacteria were delivered by spraying on flowers (Supplementary Fig. 3c) or by a bee-vectoring device con-taining SP6C4 attached to the honeybee hive (Supplementary Fig. 3d). For spraying, strain SP6C4 was grown from a spore stock for 10 days as described above and 150 ml of the bacteria ($10^9$ cfu/mL) was mixed in water to 15 L (final cell density, $10^7$ cfu/mL). In a single greenhouse, 3 control blocks and 3 sprayed blocks, each with 450 plants, were established in a completely randomized design. The bacterial mixture was applied to the flowers with a sprayer nebulizer at 1.5 L per min at 2-week intervals from sampling stage week 2 to week 8. For microbial community analysis, flower samples were harvested from January 2016 to March 2016 and pyrosequencing was carried out as described above.

For bee-vectoring, bacteria grown with shaking in PDK broth at 28 °C for 7 days were diluted to $10^7$ cfu/mL and PEG and skim milk were added to give final concentrations of 12.5% and 3%, respectively. The mixture was applied by pouring into a bee-vectoring device and attached at the front of the hive (Supplementary Fig. 3d). Disease incidence throughout the growing season was performed as described above and the replications were analyzed by ANOVA followed by Tukey's HSD test ($P = 0.05$) for mean separation.

**Entomopathogen inhibition assay**. The entomopathogens *P. larvae* and *S. mar-cescens* were grown with shaking in TSB broth at 27 °C for 4 days. Four different treatments, each with four replicates, were established: (i) control treatment; 6 mL of PDK broth added to 3 g of pollen. (ii) Pathogen treatment; pathogen cultures were diluted with sterile water to $10^5$ cfu/mL and 3 mL was introduced into a Petri dish containing 3 mL of PDK broth and 3 g of pollen. (iii) Pathogen with SP6C4 treatment; 3 mL of diluted pathogen and 3 mL of SP6C4 ($10^6$ cfu/mL) were mixed with 3 g of pollen in a Petri dish. (iv) Treatment with SP6C4 alone; 3 mL of SP6C4

was mixed with 3 mL of PDK broth and 3 g of pollen in a Petri dish. Bumble bees (*Bombus impatiens*) usually consume pollen as principal food and survive effec-tively in small cages[45]. Ten bumble bees were released into each plastic cage, the feeding mixtures were placed in the middle of the cages, and the numbers of dead bumble bees were counted every 24 h for 5 days. Entomopathogen inhibition tests were compared using one-way ANOVA followed by Duncan's mean separation test ($P$ value < 0.05).

**Statistical analyses**. All data except for sequence analyses were analyzed using SIGMA PLOT ver. 11.0 (System Software INC., Richmond, CA, USA). ANOVA was used to demonstrate differences among mean values and graphs were visua-lized by ggplots version 3.0.1, ggplot2 version 3.2.0 and agricolae (version 1.2–9) (R package).

**Reporting summary**. Further information on research design is available in the Nature Research Reporting Summary linked to this article.

## Data availability
Pyrosequencing data for flower samples have been deposited in GenBank under SAR accession number SRP150491. All other data accession numbers are available in Supplementary Table 12. The source data underlying Figs. 1a, b, e–g, 2b–d, 3d, 4b, e, 5b, c, and 5d–f and Supplementary Figs. 4a–d are provided as a Source Data file. All other relevant data and resources are available from the authors.

## Code availability
Analyses of microbial community composition were carried out with Qiime (version 1.8.0) as described at http://qiime.org/tutorials. The source code of R for data analyses is available on GitHub at https://github.com/ekfks0125/2019_Kim.git.

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

## Acknowledgements

We thank D. Schlatter of the USDA-ARS for critical comments on bioinformatics analyses and H-J, Hong of Oxford Brooks University for providing advice on *Streptomyces* molecular tools and comments on the paper. This research was supported by the Next-Generation BioGreen 21 Program (PJ013250). USDA is an Equal Opportunity Employer and Provider.

## Author contributions

D.K., D.W., L.T., and Y.K. designed and developed the experiments. D.K., G.C., and Y.K. performed pyrosequencing analyses. D.K., G.C., T.P., and Y.K. conducted genome, bioinformatics and statistical analyses. D.K. and Y.K. carried out qRT-PCR and confocal microscope work. C.J. and Y.K. performed all the greenhouse work. D.K., D.W., L.T., T.P., and Y.K. wrote the paper.

## Competing interests

The authors declare no competing interests.

## Additional information

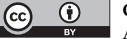

