## [Peer Review File · Nature Communications]

Reviewers' comments:

Reviewer #1 (Remarks to the Author):

The manuscript by Kim et al. identifies a plant-associated *Streptomyces* strain that can provide protection against pathogen for both its strawberry host, and a pollinator of the plant host. Further, the authors present evidence that this bacterium can move through the plant xylem, and that it can be moved by pollinators between plants. In light of these results, I think the authors describe a fascinating system where complex inter-kingdom ecological interactions can be studied. However, I find that some of the conclusions from the paper are not fully supported by the data; further, I find that the authors use a mixture of inconsistent and inappropriate ecological concepts that must be revised.

General comments

1. The authors refer to "territorial expansion" at multiple points (including the title). However, at no point the territory of any of the organisms is measured. Further, it seems like the authors are studying the *Streptomyces*' dispersion capability, not its territory. Please revise the text and conclusions under this fundamental community ecology concept, instead of the ill-defined "territorial expansion".
2. The term probiotic is used through the manuscript to refer to the *Streptomyces* strain that the authors study. I find that the concept of probiotic is not appropriate here. First, the concept is much older than what the authors claim in lines 50-51, going back to Metchnikov. See Fuller 1989 for an excellent early review. Second, the modern concept of probiotic is mostly associated with microbial treatments to a host (eg. the WHO recommends the following definition "Live microorganisms which when administered in adequate amounts confer a health benefit on the host"); however, the authors are arguably studying a naturally occurring interaction that is best captured by the classical ecology symbiosis continuum, with the interactions they study falling somewhere along the commensal-mutualist points. I would encourage the authors to reconsider their choice of concepts, and to re-interpret their results in light of a more appropriate framework.
3. The concept of 'holobiont' is brought up by the authors multiple times. This concept postulates that the interactions between a host and its associated microbes represent a distinct evolutionary unit, that undergoes selective pressures that are different from the selective pressures that influence the individual members. I fail to see where is the evidence that the tripartite (bacterium, plant, pollinator) cannot be explained by selection acting individually on each member; after all, the authors are proposing that all members benefit, therefore species level selection can explain the observations. Another hallmark of an holobiont is vertical transmission, but no evidence is presented in that regard. The authors should be clear on how their data supports the holobiont interpretation or refrain from bringing up an unnecessary concept.
4. The concept of "core microbiome" has always been operational. However, it always relies on highly replicated sampling to identify statistically prevalent taxa. The authors perform very little replication, and a lot of the biological replication is masked by technical variation given their pooling strategy. Therefore it is impossible to claim that any of the bacteria they detect is a member of the 'core microbiome'. Please refrain from using that concept or provide a quantitative and statistically justified definition of 'core'.
5. I think the way the results of the survival experiments are presented (starting line 93) is a bit confusing. It is not clear if disease was a treatment or if the plants were just naturally infected. I think the authors should dedicate a couple sentences to describe the experimental setup in the main text before describing the results.
6. I find it very cool that the authors identified the disease antagonist *Streptomyces* essentially as an anti-correlation with disease symptoms. This approach has been used before but I have never seen it being so successful. However, it seems like this bacterium is essentially out-competed by the pathogen which depletes the bacteria as it causes host damage. I wonder if this limits the potential applications of these strains and this approach. Aren't strains that are able to remain abundant (or increase their abundance) in plants inoculated with pathogen, but that remain healthy, more interesting?

7. I think the whole section about the bacterium movement in plant needs a bit of a re-write. I found it hard to follow (see a few specific comments below). Also, I wasn't clear if these experiments were performed under axenic conditions. Finally, while there is strong evidence that this bacterium can move through the xylem, I think the authors need to tone down their conclusion a bit and revisit their main interpretation. First, there is no evidence that the bacterium can move from the rhizosphere to the flowers, only to the stem, was this never tested? Second, there is movement from flowers to root, but the authors do not conclusively demonstrate that this happens through the xylem. The bacterium could move through the plant surface, or it could be that some bacterial cells fall to the soil, where they colonize the soil, then the root and then move up to the stem.

8. Unless I misunderstood it seems like a single replicate population of each treatment was performed for the gray mold disease suppression. Moreover, it seems like each replicate was carried out in a separate greenhouse (i.e. environment). Therefore, it is impossible to determine if the differences observed in disease incidence are due to technical variation, drift, environmental differences or the pathogen treatments themselves. Without replication, I don't think these results are interpretable.

9. The authors test the protective capabilities of the *Streptomyces* strain by mixing it with pathogenic bacteria and feeding the mixture to bees. I wonder how realistic this setup is. Are the pollinator pathogens typically found in pollen? They didn't seem to be in the survey data presented by the authors. Would a setup where pollinators are first challenged with the *Streptomyces* and then the pathogen be more realistic? This is not my area of expertise, but I wonder if there is any evidence that this setup is relevant for natural conditions.

10. The authors argue that they study "a unique tripartite system". While there is indeed evidence that all the three organisms are interacting, there is no data regarding the specificity of the interaction. Would other bacterial strains isolated from the flowers and pollen be able to do the same? What about other bacteria isolated from other plant organs? and other *Streptomyces* isolated from non-plant environments? Would the *Streptomyces* strain from this manuscript have the same movement and effect in other plants and pollinators? Without a specificity characterization, it is hard to argue conclusively that this system is active in nature. Therefore, the authors must be clear about the limitations of their study and tone down the conclusions accordingly.

11. Most of the last paragraph, starting around line 251 is not easy to follow. It seems like a lot of concepts that were not previously mentioned are brought up. I would encourage the authors to streamline this section to ensure it is consistent with the data and the story the present.

12. The authors do a good job describing the software and versions used. However, no code is provided. I think the authors should provide all the code and pipelines that they used. I see no reason why an exception can be made in this case.

Specific comments

1. The authors refer to their *Streptomyces* isolate as a "soil" probiotic. This is surprising since it was isolated from plant tissue. This does not mean that the plant host is its natural realized niche either. So the authors should refrain from implying that the organism's niche is known.

2. LI 27-29. While this question is interesting, the authors don't study it. The work is mostly a description of a bacterium that is somewhat capable of moving between below-ground and above-ground.

3. LI 56-57. Need a reference for the claim about flower susceptibility.

4. LI 60-61. The papers referenced show that microbes can contribute to defense but by no means that they are the "first line of defense". There is basal immunity and physical barriers that plants use for defense that act at least as early as other microbes (and probably before).

5. Since there are relatively few OTUs in the tissues sampled, I would suggest using a heatmap and clustering instead of PCA. I think that would display most of the information and they would be easier to interpret. For example, the PCAs don't show alpha-diversity which is one of the features that the authors use to characterize their samples.

6. LI 131. The term phyllosphere is used here, but anthosphere was used before. I think phyllosphere is restricted to leaves so probably anthosphere is more appropriate. In any case, be consistent and define the term on first use.

7. LI 139 I am no expert in plant tissues, but isn't there a cambium in both root and stem? To which one are you referring.
8. LI 140 and LI 143. Units are different. Be consistent.
9. LI 156. Please dedicate a couple of sentences to describe the experiment of pollinator-based movement of the bacterium.
10. Please use time units (days, weeks, etc) instead of sampling number throughout the manuscript and figures.
11. LI 166-168. Please give a few more details about this experiment in the main text (methods section is fine). For example, how many bees were used per greenhouse, what were the treatments (pathogen) applied.
12. LI 193 Earlier the authors describe the results in terms of mortality, and now they switch to survival. Please be consistent with your reported metric to facilitate reading by your audience.
13. LI 201 Are those 15 genes in one biosynthetic cluster?
14. LI 238 See my comment about territorial expansion vs dispersion, but also I don't see how air movement, bioaerosols, and insect-insect contact are related to the data presented in the manuscript.
15. LI 253. What are the costs in this system.
16. Fig 1c. Where is PC1?
17. LI 379. What does it mean that "the value was vectorized"?
18. Fig 4c. Seems like a typo "inoclated" instead of "inoculated"
19. Fig 4d-f. The authors use abundance of the phylum actinobacteria as a proxy for the abundance of their *Streptomyces* strain. It would better to use abundance of sequences that match their Strain 16S directly.
20. LI 574. The selection of a NJ algorithm is problematic since modern maximum likelihood and Bayesian methods have completely superseded NJ algorithms. Please re-create the tree with a more appropriate method.

Reviewer #2 (Remarks to the Author):

This is an exciting study that reveals that a bacterium acquired by plant roots can colonize the xylem, travel upward to flowers and be dispersed by pollinators among plants. This interaction fits clearly to the definition of a tripartite mutualism because the microbe protects both the plant and the pollinator from pathogens. Based on a previous study the authors build on this system and provide details of this interaction by (i) following the microbe along the plant and among flowers, by providing evidence on (ii) the antimicrobial action of the protective microbe, and by (iii) finding out putative genes involved in antimicrobial action. Emerging evidence suggests that many microbes protect their host plants, but how these microbes disperse among plants is little known. Finding out that dispersal is achieved using pollinators, and that these pollinators are rewarded via protection against their own pathogens is an exciting novelty. The manuscript is well written, fits the scope of the journal and will be of interest to its readers.

Despite these merits, I have few concerns that I believe the authors should address (or clarify) before publication.

1. In the section "S. globisporus SP6C4 movement in planta" the authors inoculate plants with the bacterium and follow it along the plant with fluorescent microscopy. The authors clearly show its presence in different parts of the plant. I think, however, that a negative control in which the bacterium was not inoculated is missing. This negative control should rule out that this bacterium is always present in plants, for example via vertical acquisition through seeds.

2. The authors have demonstrated microbial movement from soil to flowers, and from flowers to flowers, but they do not really connect these two experiments. The authors inoculate the soil with

the bacterium and follow its movement along the plant. Then they inoculate flowers with the bacterium and provide evidence that they are transported to new flowers by pollinators. I miss, however, that the same plants in which microbes were inoculated in the soil (experiment 1) are used to show microbial movement from flower to flower (experiment 2). Maybe an extra experiment is not needed, but a more clear explanation is needed to prove that bacteria densities in flowers in the first experiment are enough to allow their transport to new flowers.

3. Not sure I understand figure 4b. In blocked flowers, were bacteria also inoculated? If not, why are there bacteria on these plants?

4. I am not really familiar with the technical details involving mutagenesis using CRISPR/Cas9, so I can't comment much on this part.

Enric Frago,
CIRAD Montpellier.

Reviewer #3 (Remarks to the Author):

This study describes occurrence of the bacteria *Streptomyces* in strawberry flowers, over the season, how it is transmitted by bees to other flowers, and how it serves as a probiotic for the plant, as the severity of the disease *Botrytis* was reduced by streptomycis but also the pollinators benefited from the bacteria since it reduced mortality due to entomopathogens.

Overall this is a very interesting study and certainly deserves to be published in my opinion. It is very complete and the subparts build a great story.

It is important to note that, in my opinion, many parts of this story are known already. A major aspect in this story is that bees transmit the streptomyces from flower to flower. That microbes from flowers and nectar are transferred between plants is known and described (e.g. see work of Tadashi Fukami). The authors also reported this recently as the main message in another paper: Comparative tomato flower and pollinator hive microbial communities, *Journal of Plant Diseases and Protection*, 2018. Also, transfer from rhizosphere, to root to stem and flowers of rhizosphere bacteria is well described. Having said that, I remain that the story of the current manuscript is so complete and beautifully described that I certainly recommend that it is published in *Nature Communications*.

One aspect that I miss in the story is the soil compartment. Even though it is mentioned in the introduction, almost all results are about transmission of streptomyces from plant to plant by insects, occurrence in the flowers etc. In one experiment the roots and flowers are inoculated and transmission through the plant is followed from root to shoot and shoot to root.

Therefore a number of questions remain unanswered in my opinion. What is the occurrence of streptomyces in the soil, and in the rhizosphere. How does the decline of the streptomyces over time, correlating with the increase in disease severity, links to the presence of streptomyces in the rhizosphere? Why is streptomyces declining over time. This has not been addressed at all. Also in a first screening *Streptomyces* is detected in greenhouses (figure 1) but in a later experiment, with different greenhouses, there is no streptomyces in the control greenhouse and in the netted area. How can this happen. Was it present in the soil, was a fungicide used, these issues are not addressed at least I did not read it. Clearly, more information about interactions between streptomyces and root/rhizosphere could have clarified this.

Since almost all information is about the aboveground part of the interactions, the statement in

the title that this is a soil probiotic is not justified i think, based on the presented data.

The replication in the experiment with different greenhouses is not presented, as it is described as a control greenhouse and a greenhouse with nettings in part of it, I suppose there is only one of each. That means that strictly speaking the experiment is not replicated. In theory other differences between the two greenhouses than the streptomyces inoculation could have caused the effects.

Finally, I don't understand the sentence in the discussion: line 245-248: this study exemplify Commoner's laws of ecology; that "everything is connected to everything else" within an ecosphere, with members interdependent such that "there is no such thing as a free lunch." For every profit there is a cost, with all duties eventually paid to other members of the system.

This is an odd statement in this context. The study is about symbiosis as it written in the next line in the discussion, so i don't understand why the authors argue that there is no free lunch in this context. For every profit there is a cost, what do the authors mean with that in this context, while the paper shows the benefits, not the costs. I think this is really out of place.

The legend of figure 1 is incomplete and seems to come from an earlier version, c and d are not described (one pollen and one flower) and the same holds for f and g

Authors' response to the referee's comments

First of all, we appreciate your kind review and valuable comments on our manuscript. We addressed your comments in this revised manuscript as best we could and we believe that the comments and suggestions have made our manuscript stronger.

Reviewer #1 (Remarks to the Author):

The manuscript by Kim et al. identifies a plant-associated *Streptomyces* strain that can provide protection against pathogen for both its strawberry host, and a pollinator of the plant host. Further, the authors present evidence that this bacterium can move through the plant xylem, and that it can be moved by pollinators between plants. In light of these results, I think the authors describe a fascinating system where complex inter-kingdom ecological interactions can be studied. However, I find that some of the conclusions from the paper are not fully supported by the data; further, I find that the authors use a mixture of inconsistent and inappropriate ecological concepts that must be revised.

Thank you so much for your kind review and valuable comments.

General comments

1. The authors refer to "territorial expansion" at multiple points (including the title). However, at no point the territory of any of the organisms is measured. Further, it seems like the authors are studying the *Streptomyces*' dispersion capability, not its territory. Please revise the text and conclusions under this fundamental community ecology concept, instead of the ill-defined "territorial expansion".

To clarify "territorial expansion," we removed and we used more appropriate terms in both the title and conclusion sections. The title was revised as "Streptomyces, a mutualistic microbe, utilizes plant and pollinator transport for efficient dispersion."

2. The term probiotic is used through the manuscript to refer to the *Streptomyces* strain that the authors study. I find that the concept of probiotic is not appropriate here. First, the concept is much older than what the authors claim in lines 50-51, going back to Metchnikov. See Fuller 1989 for an excellent early review. Second, the modern concept of probiotic is mostly associated with microbial treatments to a host (eg. the WHO recommends the following definition "Live microorganisms which when administered in adequate amounts confer a health benefit on the host"); however, the authors are arguably studying a naturally occurring interaction that is best captured by the classical ecology symbiosis continuum, with the interactions they study falling somewhere along the commensal-mutualist points. I would encourage the authors to reconsider their choice of concepts, and to re-interpret their results in light of a more appropriate framework.

Dr. Caroline Ash, a senior editor of Science, introduced our previous paper (Cha et al. 2016. ISME J) as an editor's choice in Science. Dr. Ash used the terminology "plant probiotic" to refer to our strain. As you mention, Dr. Fuller's excellent review paper describes "probiotic" in the context of the animal or human model. However, the term "plant probiotic" is widely accepted and used fairly often in agronomy and related scientific disciplines. For example, the title of a recent paper by Menendez and Garcia-Fraile is "Plant probiotic

bacteria: solutions to feed the world.” As the reviewer notes, the origin of the strain in this study is soil/plant tissue and the strain is naturally adapted as a mutualistic microbe that is beneficial to the host. The latter part of our manuscript presents evidence that when introduced artificially, our strain contributes a health benefit to the host. However, we have toned down some of the sentences in the manuscript as you have indicated. All mention of “plant probiotic” was replaced as “mutualistic microbe”. And we agree that the three-way relationship among Streptomyces-plant-pollinator in this manuscript is more suitably described “a tripartite mutualism”. In the revised manuscript we have reduced and replaced inappropriate terms to clarify the message of our findings.

3. The concept of ‘holobiont’ is brought up by the authors multiple times. This concept postulates that the interactions between a host and its associated microbes represent a distinct evolutionary unit, that undergoes selective pressures that are different from the selective pressures that influence the individual members. I fail to see where is the evidence that the tripartite (bacterium, plant, pollinator) cannot be explained by selection acting individually on each member; after all, the authors are proposing that all members benefit, therefore species level selection can explain the observations. Another hallmark of an holobiont is vertical transmission, but no evidence is presented in that regard. The authors should be clear on how their data supports the holobiont interpretation or refrain from bringing up an unnecessary concept.

We used the term “holobiont” to introduce the idea that the plant in this study has an exceptionally strong, multifaceted relationship with a particular associated microbe. However, to avoid using inappropriate terminology we have removed the word “holobiont” in the discussion section and replaced it with terms such as “beneficial microbe,” “mutualist,” or “phytobiome” according to usage.

4. The concept of ‘core microbiome’ has always been operational. However, it always relies on highly replicated sampling to identify statistically prevalent taxa. The authors perform very little replication, and a lot of the biological replication is masked by technical variation given their pooling strategy. Therefore it is impossible to claim that any of the bacteria they detect is a member of the ‘core microbiome’. Please refrain from using that concept or provide a quantitative and statistically justified definition of ‘core’.

To use more precise terminology, we deleted “core microbiome”, “core probiotic”, “core agent” and replaced the terms with “mutualistic microbe” or “mutualistic agent” in the entire revised manuscript.

5. I think the way the results of the survival experiments are presented (starting line 93) is a bit confusing. It is not clear if disease was a treatment or if the plants were just naturally infected. I think the authors should dedicate a couple sentences to describe the experimental setup in the main text before describing the results.

Grey mold disease of strawberry always occurs and progresses naturally over time in commercial greenhouses. As we present in Fig S2, the spores of the pathogen are always present in the greenhouse and initiate infection from the weakest tissue of the plant. We added the following sentence prior to stating the results: “In most cases, the gray mold pathogen is endemic in commercial greenhouses, where it infects and causes gray mold disease of strawberry.”

6. I find it very cool that the authors identified the disease antagonist *Streptomyces* essentially as an anti-correlation with disease symptoms. This approach has been used before but I have never seen it being so successful. However, it seems like this bacterium is essentially out-competed by the pathogen which depletes the bacteria as it causes host damage. I wonder if this limits the potential applications of these strains and this approach. Aren't strains that are able to remain abundant (or increase their abundance) in plants inoculated with pathogen, but that remain healthy, more interesting?

As presented in Fig 4d and 4e in this revised manuscript, when the strain is sprayed in the greenhouse, gray mold disease occurrence is successfully suppressed by the SP6C4. This suggests the bacteria are not out-competed by the gray mold pathogen. Additionally, when sprayed in the greenhouse, Streptomyces OTUs increase and remain abundant in flower tissue (Fig S5). Why do plants not support an effective Streptomyces population density throughout their entire life time under natural conditions? We believe that it may depend on the plant aging. Fresh flowers of strawberry bloom through three periods from November to April, but the plant itself is getting older. With aging, the plant metabolism and exudate composition may change. Biochemicals with a critical role in supporting Streptomyces population density on flowers may become less available upon aging of the plant. Currently, we are performing experiments to reveal possible correlations between the strain population density and plant exudate composition. The results will be submitted in a later manuscript.

7. I think the whole section about the bacterium movement in plant needs a bit of a re-write. I found it hard to follow (see a few specific comments below). Also, I wasn't clear if these experiments were performed under axenic conditions. Finally, while there is strong evidence that this bacterium can move through the xylem, I think the authors need to tone down their conclusion a bit and revisit their main interpretation. First, there is no evidence that the bacterium can move from the rhizosphere to the flowers, only to the stem, was this never tested? Second, there is movement from flowers to root, but the authors do not conclusively demonstrate that this happens through the xylem. The bacterium could move through the plant surface, or it could be that some bacterial cells fall to the soil, where they colonize the soil, then the root and then move up to the stem.

All movement experiments were performed in commercial potting mix because it provides conditions similar to those in commercial greenhouses. The growth conditions and soil were as described in the methods section and the pots and soil surfaces were covered with plastic wrap to prevent bacterial spread from flower inoculation sites to soil.

*The part about movement was re-written as follows: Identity among the three strains also suggested that they could colonize flowers and even pollen attached to bee bodies, leading us to hypothesize that the bacteria establish in the rhizosphere, move upward internally as endophytes, and can transfer to pollinators. This model is consistent with a mutualistic relationship in an ecosystem that includes both plants and insects. The hypothesis was tested in two separate treatments conducted by either dipping roots of strawberry into a suspension of strain SP6C4 (10^6 cfu/mL) or by dropping 100 μ L of the bacterial culture by pipette onto flower stamens and carpels. Both treatments included two independent experiments, one in which the introduced strain was detected with the strain specific marker gene, *lanM*²⁵ and the other in which it was labeled with mCherry. All stem, crown and root tissues were surface-sterilized prior to fluorescence microscopy and analysis by qPCR for *lanM*. Five days after root dipping, the bacteria were detected in the endosphere of root tissue by qPCR (Fig. 3a,b). By 10 days, mCherry fluorescence was seen*

in root and stem vascular bundle tissues (Fig. 3c) which, like those harboring the wild-type strain, tested positive for lanM and maintained population sizes of up to 10⁶ lanM copy/g of tissue until 20 days (Fig. 3b,c). At 30 days, both mCherry fluorescence and population sizes tested by qPCR were greatly reduced in the xylem of the stem (Fig. 3d).

After flower inoculation, the mCherry-labeled strain was observed on the stigma and pistil of flowers (Fig. 3f,h). The bacteria retained a population size of at least 10⁵ lanM copy/g of tissue and mCherry fluorescence was observed along the filament tissues until 15 days (Fig. 3e,g,i). In cross sections just above the crown, the middle part of the stem and the petiole, mCherry fluorescence was detected in vascular bundles from 10 days (Fig. 3j) up to 30 days (Fig. 3k). The bacteria were initially detected up to 10⁵ lanM copy/g of tissue in both the stem at 10 days and the crown at 15 days, where high population densities were maintained for up to 30 days (Fig. 3e). In cross sections just above the crown tissue, middle part of the stem and the petiole, fluorescence was detected from 10 days (Fig. 3j) up to 30 days (Fig. 3k). mCherry fluorescence was observed in vascular bundles of the tissues. These results provide evidence that SP6C4 can translocate endophytically from the rhizosphere to above-ground plant tissues, and from the anthosphere to the root.

We tried many times to detect bacterial movement from the rhizosphere to the flower. We set up pre-budding plants and introduced the bacteria on the root. Unfortunately, we never succeeded in obtaining flowering plants over a period of 40 days under lab conditions. We recognize that this is not a good excuse, but we believe that the results, the microbe movement by pollinator from flower to flower may provide enough evidence of bacterial movement and its interaction with plant and pollinator. Therefore, we hope the reviewer will accept our experimental design and results.

8. Unless I misunderstood it seems like a single replicate population of each treatment was performed for the gray mold disease suppression. Moreover, it seems like each replicate was carried out in a separate greenhouse (i.e. environment). Therefore, it is impossible to determine if the differences observed in disease incidence are due to technical variation, drift, environmental differences or the pathogen treatments themselves. Without replication, I don't think these results are interpretable.

Please see our comments on the same questions from the editor (above).

9. The authors test the protective capabilities of the Streptomyces strain by mixing it with pathogenic bacteria and feeding the mixture to bees. I wonder how realistic this setup is. Are the pollinator pathogens typically found in pollen? They didn't seem to be in the survey data presented by the authors. Would a setup where pollinators are first challenged with the Streptomyces and then the pathogen be more realistic? This is not my area of expertise, but I wonder if there is any evidence that this setup is relevant for natural conditions.

In nature, entomopathogens usually contaminate or colonize insect food such as pollen and the pathogens infect the insect when the insect eats the contaminated pollen. The pathogens cause a deadly problem in the intestine of the insect. The pollinators were introduced into a greenhouse with their hive and pollen as a food source. If the SP6C4 was fed before exposure to the pathogen, the bee mortality might be more significantly reduced than if fed with both microorganisms at the same time. However, we are not sure which microbes might colonize the pollen first, therefore we set up the experiment as feeding both

microbes at the same time. We believe this treatment has no critical issue for this experiment.

10. The authors argue that they study “a unique tripartite system”. While there is indeed evidence that all the three organisms are interacting, there is no data regarding the specificity of the interaction. Would other bacterial strains isolated from the flowers and pollen be able to do the same? What about other bacteria isolated from other plant organs? and other *Streptomyces* isolated from non-plant environments? Would the *Streptomyces* strain from this manuscript have the same movement and effect in other plants and pollinators? Without a specificity characterization, it is hard to argue conclusively that this system is active in nature. Therefore, the authors must be clear about the limitations of their study and tone down the conclusions accordingly.

SP6C4 showed outstanding ability to colonize the flower, bee body and even the gut of the pollinator (Fig S4). Without a specific interaction, artificially introduced microbes cannot maintain their population density on the plant or insect. There are many previous studies that have demonstrated this phenomenon. For example, certain strains of Pseudomonas spp. (diacetylphloroglucinol or phenazine antibiotic producers) colonize plant roots with high cell density and maintain enriched population size over time in the wheat rhizosphere. A special interaction between the Pseudomonas-wheat plant relationship supports this relationship. Streptomyces SP6C4, SF7B6 and S4-7 were originally isolated from different tissues of the plant at different times and locations, but the strains show the same biological ability to interact with strawberry. The data do not rule out that other microbes may also have a special interaction with the strawberry plant. However, based on our previous study (Kim et al. 2019. MPMI), Streptomyces spp., isolated from other plants or non-cultivated soil do not have this specific interaction with the strawberry plant and fail to colonize the root of strawberry. Therefore, we believe the tripartite mutualism in this study, the relationship among Streptomyces SP6C4-Strawberry-Bee, is “novel”.

11. Most of the last paragraph, starting around line 251 is not easy to follow. It seems like a lot of concepts that were not previously mentioned are brought up. I would encourage the authors to streamline this section to ensure it is consistent with the data and the story the present.

As per your comment, we eliminated, simplified, and re-organized the section to avoid unnecessary argument. “To better understand the persistence of microorganisms in the environment requires detailed knowledge of their individual capabilities and needs as well as recognition of the give-and-take involved in their interactions with other members of the ecosystem. The relationships among the microbe, the plant, and the insect in this study exemplify Commoner’s laws of ecology³⁴; that “everything is connected to everything else” within an interaction, with members interdependent. In our study, the plant served as a host in the space inhabited by the three partners, the insect provided plant-to-plant transport and the Streptomyces maintained the partnership by protecting against pathogens that attack the plant and the insect. From an ecological perspective, when the soil, the plants and the pollinators are considered as separate environments inhabited by microorganisms, the Streptomyces in our system transformed the three into a mutualistic continuum with the plants. Our findings present mutualistic interactions among individual members, but natural or agricultural ecosystems have many more complex and sophisticated elements. A better understanding of the relationships within such mutualistic phytobiomes will facilitate the use of the native microbiota and specific introduced agents to defend against biotic and abiotic stresses, thus more sustainably protecting the world’s food supply.”

12. The authors do a good job describing the software and versions used. However, no code is provided. I think the authors should provide all the code and pipelines that they used. I see no reason why an exception can be made in this case.

All codes have been submitted in [github] and we provide a URL in the revised manuscript.

Specific comments

1. The authors refer to their *Streptomyces* isolate as a “soil” probiotic. This is surprising since it was isolated from plant tissue. This does not mean that the plant host is its natural realized niche either. So the authors should refrain from implying that the organism’s niche is known.

SP6C4 was isolated from strawberry flowers at sampling stage “6 weeks”, SF7B6 was obtained from the pollen at sampling stage “7 weeks”. The two strains were isolated from different locations within a 2-week interval. Strain S4-7 was isolated from the strawberry rhizosphere 2 years before SP6C4 and SFB6, from a strawberry field approximately 40 kilometers away from where SP6C4 and SFB6 were recovered. Given that streptomycetes are common soil inhabitants, we consider it appropriate to describe the soil as their natural habitat.

2. LI 27-29. While this question is interesting, the authors don’t study it. The work is mostly a description of a bacterium that is somewhat capable of moving between below-ground and above-ground.

Plants always have interactions with their surrounding environmental factors, including microbes, and microorganisms also have a strong relationship with plants. Some interactions between plants and microbes are deleterious, such as when pathogens invade the host plant. Some microbe-plant relationships are beneficial to both organisms. LI 27-29 is the first sentence of the abstract and “instruction to authors” suggests that the first statement should start with general information.

3. LI 56-57. Need a reference for the claim about flower susceptibility.

*Added a reference “Suzuki, N. et al. Abiotic and biotic stress combination. New Phytol. **203**, 32-43. (2014).”*

4. II. 60-61. The papers referenced show that microbes can contribute to defense but by no means that they are the “first line of defense”. There is basal immunity and physical barriers that plants use for defense that act at least as early as other microbes (and probably before).

We agree with the reviewers’ opinion. Physical barriers and basal immunity activity of plants are an important initial defense system, there is no doubt. However, many microbe-microbe interactions occur on plant surfaces prior to breaching plant physical barriers, and it is not unusual for plants to lack resistance or immunity responses, especially against economically important soil-borne necrotrophic pathogens. There are many examples in which plants develop a mutualistic relationship with microbes to suppress pathogen activity or disease development. Therefore, the current statements from previous reports should be kept in the manuscript.

5. Since there are relatively few OTUs in the tissues sampled, I would suggest using a heatmap and clustering instead of PCA. I think that would display most of the information and they would be easier to interpret. For example, the PCAs don't show alpha-diversity which is one of the features that the authors use to characterize their samples.

As reviewers' comment, we replaced PCAs with heatmaps with hierarchical clustering.

6. LI 131. The term phyllosphere is used here, but anthosphere was used before. I think phyllosphere is restricted to leaves so probably anthosphere is more appropriate. In any case, be consistent and define the term on first use.

We carefully checked the word "phyllosphere" in the sentences and replaced the word with "anthosphere" or "above-ground tissue" according to usage.

7. LI 139 I am no expert in plant tissues, but isn't there a cambium in both root and stem? To which one are you referring.

As we understand, the strawberry stem has cambium tissue, from which eventually develops the vascular bundles. Cambium tissue in strawberry root is very difficult to observe because the vascular tissue is fully developed at an early stage of root development. To make the statement clear, the sentence was modified: "By 10 days, mCherry fluorescence was seen in root and stem vascular bundle tissues (Fig. 3c) which, like those harboring the wild-type strain, tested positive for lanM and maintained population sizes of up to 10^6 lanM copy/g of tissue until 20 days (Fig. 3b,c)"

8. LI 140 and LI 143. Units are different. Be consistent.

Fixed

9. LI 156. Please dedicate a couple of sentences to describe the experiment of pollinator-based movement of the bacterium.

We have added the following sentences to describe the experiment: "To further evaluate movement from plant to plant by a pollinator, three sets of flowers were prepared in a cage. The first set of flowers received suspensions of a hygromycin-marked strain (10^6 cfu/mL) on the surface of flower stamens and carpels, the second set received no bacterial treatment and the third set of flowers also had no bacteria and was covered with a cap to prevent honeybee access (Fig. 4a)."

10. Please use time units (days, weeks, etc) instead of sampling number throughout the manuscript and figures.

Units were replaced as intervals of sampling, i.e., 2 week, 4 week, 6 week,...

11. LI 166-168. Please give a few more details about this experiment in the main text (methods section is fine). For example, how many bees were used per greenhouse, what were the treatments (pathogen) applied.

In general, 10,000 to 20,000 honeybees live in a single wooden beehive but we are not sure exactly how many bees existed in the hive we used. Only a single hive was introduced per greenhouse (approximately 84 meter long X 8 meter wide) for pollination. In our response to comment No.5 above, we mentioned that gray mold disease was caused naturally by the indigenous spore population. We added these details to the methods section.

12. II. 193 Earlier the authors describe the results in terms of mortality, and now they switch to survival. Please be consistent with your reported metric to facilitate reading by your audience.

We have used “mortality” throughout in the revised manuscript to be consistent in wording.

13. LI 201 Are those 15 genes in one biosynthetic cluster?

As per the editor’s comment, this part was eliminated from the manuscript. However, the SP6C4 strains have a total of 15 putative antibiotic biosynthetic gene clusters and all 15 clusters are independent in SP6C4 genome.

14. LI 238 See my comment about territorial expansion vs dispersion, but also I don’t see how air movement, bioaerosols, and insect-insect contact are related to the data presented in the manuscript.

To be toned down, the sentences were modified as follows “Our evidence that SP6C4 is transferred from the rhizosphere to above-ground as well as from flower to pollinator that offers adapted microflora not only the potential for dispersion, but also efficient translocation to other plants that subsequently benefit as the Streptomyces move systemically throughout them.”

15. LI 253. What are the costs in this system.

To make the concept consistent and easy to follow for readers, this part has been removed from the Discussion in line with your comments above. However, “the costs” refer to the metabolic costs to Streptomyces in producing the antibiotics that provide protection against pathogens for both the plant and bee.

Questions 16-17: the below parts were removed in this revised manuscript according to the editor’s comment.

16. Fig 1c. Where is PC1?

17. LI 379. What does it mean that “the value was vectorized”?

As per your comment, the PCoA figures were deleted and replaced by heatmap figures.

18. Fig 4c. Seems like a typo “inoclated” instead of “inoculated”

Fixed

19. Fig 4d-f. The authors use abundance of the phylum actinobacteria as a proxy for the abundance of their *Streptomyces* strain. It would better to use abundance of sequences that match their Strain 16S directly.

Fig 4 d-f were replaced with new data sets. Also, Supplementary Fig 5c,d show abundance of the LanM gene, a more specific gene to detect the SP6C4 strain (Please see Kim et al. MPMI 2019, in the references, and the information was mentioned in the results).

20. LI 574. The selection of a NJ algorithm is problematic since modern maximum likelihood and Bayesian methods have completely superseded NJ algorithms. Please re-create the tree with a more appropriate method.

The phylogenetic tree was re-created with Maximum likelihood and Bayesian methods.

Reviewer #2 (Remarks to the Author):

This is an exciting study that reveals that a bacterium acquired by plant roots can colonize the xylem, travel upward to flowers and be dispersed by pollinators among plants. This interaction fits clearly to the definition of a tripartite mutualism because the microbe protects both the plant and the pollinator from pathogens. Based on a previous study the authors build on this system and provide details of this interaction by (i) following the microbe along the plant and among flowers, by providing evidence on (ii) the antimicrobial action of the protective microbe, and by (iii) finding out putative genes involved in antimicrobial action. Emerging evidence suggests that many microbes protect their host plants, but how these microbes disperse among plants is little known. Finding out that dispersal is achieved using pollinators, and that these pollinators are rewarded via protection against their own pathogens is an exciting novelty. The manuscript is well written, fits the scope of the journal and will be of interest to its readers.

Despite these merits, I have few concerns that I believe the authors should address (or clarify) before publication.

Thank you so much for your kind review and valuable comments.

1. In the section "S. globisporus SP6C4 movement in planta" the authors inoculate plants with the bacterium and follow it along the plant with fluorescent microscopy. The authors clearly show its presence in different parts of the plant. I think, however, that a negative control in which the bacterium was not inoculated is missing. This negative control should rule out that this bacterium is always present in plants, for example via vertical acquisition through seeds.

In Fig 3, mCherry fluorescent microscopy images showed negative controls, which were untreated plant tissues. Please see the right side panel of Fig 3 c, d, f, g, h, i, j, and k.

2. The authors have demonstrated microbial movement from soil to flowers, and from flowers to flowers, but they do not really connect these two experiments. The authors

inoculate the soil with the bacterium and follow its movement along the plant. Then they inoculate flowers with the bacterium and provide evidence that they are transported to new flowers by pollinators. I miss, however, that the same plants in which microbes were inoculated in the soil (experiment 1) are used to show microbial movement from flower to flower (experiment 2). Maybe an extra experiment is not needed, but a more clear explanation is needed to prove that bacteria densities in flowers in the first experiment are enough to allow their transport to new flowers.

We agree with the comments. Translocation from rhizosphere to above-ground (exp 1), from flower to other flower (exp 2) and from flower to below-ground (exp 3) used different plants and were performed as individual experiments. This may be considered a weakness or limitation of the manuscript. However, as the reviewer has said, the results of each experiment present reliable data showing that the strain can move from below-ground to above ground (Fig 3b, c, d), from flower to another flower by pollinator (Fig 4a, b), and from flower to below-ground (Fig 3e, f, g, h, i, j, and K). The strain densities in each tissue also are sufficient to allow such movement. Fig 3b shows that the strain moves from rhizosphere to above-ground at 10^6 cfu/g, Fig 4b shows that the strain was translocated to uninoculated flowers at 10^6 cfu/g, and finally, Fig 3e shows 10^6 cfu/g of bacterial density in the rhizosphere after movement from the flower.

3. Not sure I understand figure 4b. In blocked flowers, were bacteria also inoculated? If not, why are there bacteria on these plants?

Fig 4b has been replaced with more precise data and new supplementary data were added (Fig S5). Fig 4b shows three treatments and four results of the strain densities. First, only the inoculated flowers received the bacterial strain at 10^6 cfu/ml density. Non-inoculated flowers (blocked from bee visiting by a cap) showed no bacterial density on hygromycin media (80 ug/mL). Bee bodies had the strain at 10^4 cfu/ml and non-inoculated (allowed the bee visiting) flower presented the bacteria at a density at 10^5 cfu/ml. The new supplementary data show that the strain can even be detected on the body of bees as well as the gut.

4. I am not really familiar with the technical details involving mutagenesis using CRISPR/Cas9, so I can't comment much on this part.

In accordance with the editor's comment, the mutagenesis parts (result, discussion and methods) using CRISPR/Cas9 were removed in this revised manuscript.

Reviewer #3 (Remarks to the Author):

This study describes occurrence of the bacteria *Streptomyces* in strawberry flowers, over the season, how it is transmitted by bees to other flowers, and how it serves as a probiotic for the plant, as the severity of the disease *Botrytis* was reduced by *streptomyces* but also the pollinators benefited from the bacteria since it reduced mortality due to entomopathogens. Overall this is a very interesting study and certainly deserves to be published in my opinion. It is very complete and the subparts build a great story. It is important to note that, in my opinion, many parts of this story are known already. A major aspect in this story is that bees transmit the *streptomyces* from flower to flower. That microbes from flowers and nectar are transferred between plants is known and described (e.g. see work of Tadashi Fukami). The authors also reported this recently as the main message in another paper: Comparative

tomato flower and pollinator hive microbial communities, *Journal of Plant Diseases and Protection*, 2018. Also, transfer from rhizosphere, to root to stem and flowers of rhizosphere bacteria is well described. Having said that, I remain that the story of the current manuscript is so complete and beautifully described that I certainly recommend that it is published in *Nature Communications*.

Thank you so much for your kind review and valuable comments.

One aspect that I miss in the story is the soil compartment. Even though it is mentioned in the introduction, almost all results are about transmission of streptomycetes from plant to plant by insects, occurrence in the flowers etc. In one experiment the roots and flowers are inoculated and transmission through the plant is followed from root to shoot and shoot to root. Therefore a number of questions remain unanswered in my opinion. What is the occurrence of streptomycetes in the soil, and in the rhizosphere. How does the decline of the streptomycetes over time, correlating with the increase in disease severity, links to the presence of streptomycetes in the rhizosphere? Why is streptomycetes declining over time. This has not been addressed at all. Also in a first screening *Streptomyces* is detected in greenhouses (figure 1) but in a later experiment, with different greenhouses, there is no streptomycetes in the control greenhouse and in the netted area. How can this happen. Was it present in the soil, was a fungicide used, these issues are not addressed at least I did not read it. Clearly, more information about interactions between streptomycetes and root/rhizosphere could have clarified this.

Responses to the first question are linked to our previous papers (Cha et al. 2016. ISME J and Kim et al. MPMI. 2019), which are the key references for the current study. Cha et al. reported a rhizosphere strain, Streptomyces S4-7, and results showing that when the bacterium was artificially introduced into soil, it was maintained at a population density above 10^6 cfu/g of rhizosphere soil for up to nearly 7 weeks regardless of the presence of the Fusarium wilt pathogen.

We did not check the correlation between gray mold (an above-ground disease) disease incidence and the strain density in the rhizosphere. Indeed, it would be an interesting study! Our previous paper (Kim et al. MPMI. 2019) presented an inverse correlation between density of the strain and Fusarium wilt disease occurrence in 30 commercial strawberry greenhouses. The more important finding by Kim et al. was that the key microbe (a Streptomyces strain) was detected only in the strawberry rhizosphere among various plant rhizospheres, including several un-cultivated soil samples. As we responded to reviewer 1's question, decline of the strain population above ground may be due to plant aging. At the starting point (14 weeks) of the decline of bacterial density, the strawberry had the third flush of blooming flowers. We believe that at this stage, the strawberry plant has dramatically less vigor compared to the 1st and 2nd blooming periods. Some exudate components could be reduced or changed based on the aging, and the exudate might have a critical role in supporting the bacterial population density.

Since almost all information is about the aboveground part of the interactions, the statement in the title that this is a soil probiotic is not justified I think, based on the presented data.

We agree and changed the title, using "mutualistic microbe" instead of "soil probiotic" in this revised manuscript

The replication in the experiment with different greenhouses is not presented, as it is described as a control greenhouse and a greenhouse with nettings in part of it, I suppose there is only one of each. That means that strictly speaking the experiment is not replicated. In theory other differences between the two greenhouses than the streptomyces inoculation could have caused the effects.

Please, see the comments to the editor (above in document) regarding this same question

Finally, I don't understand the sentence in the discussion: line 245-248: this study exemplify Commoner's laws of ecology; that "everything is connected to everything else" within an ecosphere, with members interdependent such that "there is no such thing as a free lunch." For every profit there is a cost, with all duties eventually paid to other members of the system.

This is an odd statement in this context. The study is about symbiosis as it written in the next line in the discussion, so i don't understand why the authors argue that there is no free lunch in this context. For every profit there is a cost, what do the authors mean with that in this context, while the paper shows the benefits, not the costs. I think this is really out of place.

To tone it down and to avoid unnecessary debate, we deleted a part of the sentence and re-wrote as follows: "To better understand the persistence of microorganisms in the environment requires detailed knowledge of their individual capabilities and their needs as well as recognition of the give-and-take involved in their interactions with other members of the community. The relationships among the microbe, the plant, and the insect in this study exemplify Commoner's laws of ecology³⁵; that "everything is connected to everything else" within an interaction, with members interdependent on each other. In our study, the plant served as a host in the space inhabited by the three partners, the insect provided plant-to-plant transport and the Streptomyces maintained the partnership by producing antibiotics that benefited both the plant and the insect."

The legend of figure 1 is incomplete and seems to come from an earlier version, c and d are not described (one pollen and one flower) and the same holds for f and g

The legends of Fig 1 were fixed.

REVIEWERS' COMMENTS:

Reviewer #1 (Remarks to the Author):

I found the updated manuscript by Kim et al. much improved and easier to read. While I still have one relatively major comment (see below) I find that the authors have addressed my original major concerns, and I find the system truly fascinating. Specifically, the authors clarified the replication and methods; improved their phylogenetic analysis; provided access to code and data; changed some figures to better illustrate their results and made a big effort for improving the precision of the language throughout the text.

I list my remaining comments below:

1. I find the new title better, but it seems like the authors are implying that the whole *Streptomyces* genus is mutualistic. Just clarify that you are referring to a specific strain.
2. My main remaining comment is regarding that there is no direct demonstration of movement from the rhizosphere into the flower. I appreciate the authors clarification regarding their negative results in their rebuttal letter. These negative results do not challenge the fundamental finding of the tripartite interaction, but I wonder if the authors shouldn't re-examine the role of the rhizosphere bacteria in their working model. Pollinator-mediated flower to flower was demonstrated and the protective effect of the *Streptomyces* strain is sufficient to explain the maintenance of the system, why then is it important that the microbe moves from the rhizosphere into the flower? While this could happen in some conditions the evidence suggests to me that such event is rare at best. It seems like the whole system is self contained without reference to the rhizosphere.
3. Related to the point above, and clearly out of the scope for this manuscript, I think it would be cool if the authors could demonstrate protection against *Botrytis* (and plant to plant movement) after inoculating only the soil instead of spraying.
4. Line 107. typo should be "324 isolates were recovered".

Reviewer #2 (Remarks to the Author):

The authors of this exciting study have addressed my concerns, and I recommend this manuscript to be accepted for publication.

Enric Frago
CIRAD

Reviewer #3 (Remarks to the Author):

I have re-assessed the ms and am satisfied with most of the answers and revisions of the authors.

However, one of my major points of criticisms has not been answered sufficiently. I noted previously that while the emphasis of the text is largely of the soil-plant-insect link, there is very little evidence for the soil to plant link in the ms. most of the experiments are about transmission from one plant to the other, and about the functions of streptomycetes. transmission from plant to plant by insects has been shown recently by the same authors in other papers.

The soil link however, is barely present. The only piece of evidence comes from the root dipping experiment. This shows that streptomycetes can move to the stem, but does not provide any evidence for all the other statements that are made. I would have expected experiments with soil inoculations and then show how this

suppresses disease severity aboveground, and studies with inoculation at one location, and then studying transmission by insects and effects on other plants, that have not been exposed to soil inoculation.

As mentioned earlier, i find this a very interesting study, with an impressive amount of work done, and certainly worth publishing, but the claims about aboveground-belowground interactions are not/barely backed-up by experiments or results even though this part is central to the story.

Authors' response to the referee's comments

First of all, we appreciate your kind review and valuable comments on our manuscript. We addressed your comments in this final revised manuscript as best we could and we believe that the comments and suggestions have made our manuscript stronger and more clear.

Reviewer #1 (Remarks to the Author):

I found the updated manuscript by Kim et al. much improved and easier to read. While I still have one relatively major comment (see below) I find that the authors have addressed my original major concerns, and I find the system truly fascinating. Specifically, the authors clarified the replication and methods; improved their phylogenetic analysis; provided access to code and data; changed some figures to better illustrate their results and made a big effort for improving the precision of the language throughout the text.

>> *We appreciate your many valuable comments on our manuscript.*

1. I find the new title better, but it seems like the authors are implying that the whole *Streptomyces* genus is mutualistic. Just clarify that you are referring to a specific strain.

>> *The title has been changed to "A mutualistic interaction between *Streptomyces* bacteria, strawberry plants and pollinating bees"*

2. My main remaining comment is regarding that there is no direct demonstration of movement from the rhizosphere into the flower. I appreciate the authors clarification regarding their negative results in their rebuttal letter. These negative results do not challenge the fundamental finding of the tripartite interaction, but I wonder if the authors shouldn't re-examine the role of the rhizosphere bacteria in their working model. Pollinator-mediated flower to flower was demonstrated and the protective effect of the *Streptomyces* strain is sufficient to explain the maintenance of the system, why then is it important that the microbe moves from the rhizosphere into the flower? While this could happen in some conditions the evidence suggests to me that such event is rare at best. It seems like the whole system is self contained without reference to the rhizosphere.

>> *Thank you for your kind acceptance of our explanation in the rebuttal letter. We believe that plants need an antimicrobial microbe to protect their flowers tissue against air-borne pathogens such as *Botrytis*. Soil is the home of numerous microorganisms, and plants develop a mutualistic relationship with certain microbe(s), which can suppress pathogens, and the microbe(s) to move from the rhizosphere to the phyllosphere, including flower tissue. In this manuscript, we mentioned our previous paper (Cha et al. 2016. ISME J: Microbial and biochemical basis of a *Fusarium* wilt-suppressive soil) in the introduction and discussion. The paper dissected the Plant-*Streptomyces* interaction at the rhizosphere level.*

3. Related to the point above, and clearly out of the scope for this manuscript, I think it would be cool if the authors could demonstrate protection against *Botrytis* (and plant to plant movement) after inoculating only the soil instead of spraying.

>> *This is the great idea! We will design an experiment to test whether rhizosphere*

introduced SP6C4 can protect the flowers against gray mold or other air-borne diseases. Once again, thank you for your kind comment and suggesting the idea for a future study.

4. Line 107. typo should be “324 isolates were recovered”.

>> *fixed*

Reviewer #2 (Remarks to the Author):

The authors of this exciting study have addressed my concerns, and I recommend this manuscript to be accepted for publication.

Enric Frago

>> *Dr. Frago, Thank you so much your comments and contributions to our manuscript.*

Reviewer #3 (Remarks to the Author):

I have re-assessed the ms and am satisfied with most of the answers and revisions of the authors.

>> *Thank you for your valuable review and suggestions about our manuscript.*

However, one of my major points of criticism has not been answered sufficiently. I noted previously that while the emphasis of the text is largely of the soil-plant-insect link, there is very little evidence for the soil to plant link in the ms. most of the experiments are about transmission from one plant to the other, and about the functions of streptomyces. transmission from plant to plant by insects has been shown recently by the same authors in other papers. The soil link however, is barely present. The only piece of evidence comes from the root dipping experiment. This shows that *Streptomyces* can move to the stem, but does not provide any evidence for all the other statements that are made.

>> *With regard to the link between soil or rhizosphere, Streptomyces and the plant, we addressed this point in our previous paper (Cha et al. 2016. ISME J). The paper characterized the strawberry rhizosphere Streptomyces strain and its interaction with the plant as well as molecular and biochemical characteristics in antimicrobial and soil-born pathogen suppressiveness. Our current manuscript builds on our previous work and focuses on how and why the Streptomyces can move through the endosphere of plants and also from plant to plant.*

I would have expected experiments with soil inoculations and then show how this suppresses disease severity aboveground, and studies with inoculation at one location, and then studying transmission by insects and effects on other plants, that have not been exposed to soil inoculation. As mentioned earlier, i find this a very interesting study, with an impressive amount of work done, and certainly worth publishing, but the claims about aboveground-belowground interactions are not/barely backed-up by experiments or results even though this part is central to the story.

>> *Reviewer #1 also suggested this idea. Again, the idea is fascinating and an exciting suggestion. We will set up a new research project based on the results of our current manuscript. Once again, thank you for your valuable suggestion.*